# Combining chemical and genetic approaches to increase drought resistance in plants

Min-Jie Cao[1], Yu-Lu Zhang[1], Xue Liu[1], Huan Huang[1], X. Edward Zhou[2], Wen-Long Wang[1], Ai Zeng[1], Chun-Zhao Zhao[3], Tong Si[3], Jiamu Du [1], Wen-Wu Wu[1], Fu-Xing Wang[1], H. Eric Xu[2,4] & Jian-Kang Zhu[1,3]

Drought stress is a major threat to crop production, but effective methods to mitigate the adverse effects of drought are not available. Here, we report that adding fluorine atoms in the benzyl ring of the abscisic acid (ABA) receptor agonist AM1 optimizes its binding to ABA receptors by increasing the number of hydrogen bonds between the compound and the surrounding amino acid residues in the receptor ligand-binding pocket. The new chemicals, known as AMFs, have long-lasting effects in promoting stomatal closure and inducing the expression of stress-responsive genes. Application of AMFs or transgenic overexpression of the receptor PYL2 in *Arabidopsis* and soybean plants confers increased drought resistance. The greatest increase in drought resistance is achieved when AMFs are applied to the PYL2-overexpression transgenic plants. Our results demonstrate that the combining of potent chemicals with transgenic overexpression of an ABA receptor is very effective in helping plants combat drought stress.

[1] Shanghai Center for Plant Stress Biology and Center of Excellence in Molecular Plant Sciences, Chinese Academy of Sciences, 300 Fenglin Road, Shanghai 200032, China. [2] Laboratory of Structural Sciences, Center for Structural Biology and Drug Discovery, Van Andel Research Institute, 333 Bostwick Ave., N.E., Grand Rapids, MI 49503, USA. [3] Department of Horticulture and Landscape Architecture, Purdue University, West Lafayette, IN 47907, USA. [4] VARI-SIMM Center, Center for Structure and Function of Drug Targets, Key Laboratory of Receptor Research, Shanghai Institute of Materia Medica, Chinese Academy of Sciences, Shanghai 201203, China. Min-Jie Cao, Yu-Lu Zhang and Xue Liu contributed equally to this work. Correspondence and requests for materials should be addressed to J.-K.Z. (email: jkzhu@sibs.ac.cn)

Drought is a serious and worldwide problem for crop production[1]. Plants respond to drought stress by complex adjustments including stomatal closure and induction of drought-responsive genes, in which the phytohormone abscisic acid (ABA) plays a major role[2]. Drought stress stimulates ABA biosynthesis[3–5]. ABA binds to the PYR/PYL/RCAR (referred to as PYL hereafter) family of receptors to inhibit clade A PP2Cs (type 2C protein phosphatases), thus activating SnRK2 kinases (subfamily 2 members of SNF1-related kinases) by releasing them from inhibition by the PP2Cs[6–11]. Activated SnRK2s phosphorylate downstream effectors, such as the S-type anion channel SLAC1 and b-ZIP transcriptional factors, to trigger stomatal closure and expression of stress-responsive genes[12–15]. X-ray crystal structure analysis of PYL–ABA–PP2C complexes has further elucidated ABA perception and initial signal transduction processes[16,17]. Upon ABA binding to the pocket of PYL, a conserved tryptophan residue in the PP2C inserts into the PYL ligand-binding pocket and forms water-mediated hydrogen bonds with ABA, which locks ABA in the PYL pocket[16,17].

The PYL receptors belong to a highly conserved ligand-binding protein superfamily containing a START domain[6,7,18]. Higher plants contain multiple PYL receptors. Based on sequence similarity, 14 PYL family members occur in *Arabidopsis thaliana*, with 11 orthologs in rice, 20 in maize, and 23 in soybean. In *Arabidopsis*, PYL receptors are divided into two groups based on the oligomeric state in ligand-free forms. PYR1 and PYL1–PYL3 are homodimers, whereas PYL4–PYL10 (except the untested PYL7) are monomers in the absence of ABA[19,20]. The monomeric PYLs can bind PP2C proteins without the involvement of ABA and may mediate ABA-independent processes in addition to ABA-triggered abiotic stress resistance[20].

A critical function of ABA in drought response is to close stomata and thus reduce transpirational water loss. In addition, ABA induces many stress-responsive genes that contribute to osmotic adjustment, dehydration tolerance, management of reactive oxygen species, and other adaptive responses. Therefore, application of ABA could in principle increase drought resistance in crops. However, rapid catabolism and chemical instability of ABA have limited its application in the field[21,22]. Researchers have therefore searched for small compounds that function as ABA analogs but with increased chemical and/or physiological stability. Pyrabactin, a synthetic germination inhibitor, is the first artificial ABA analog, and it specifically binds to a subset of PYL receptors in *Arabidopsis*[6]. AM1 (ABA mimic 1)[23], also known as quinabactin[24], is later identified as a more effective pan-agonist of PYLs. AM1, by promoting PYL–PP2C binding, not only inhibits seed germination but also increases drought resistance by reducing water loss and activating stress-responsive genes. While these effects make AM1 potentially useful for agriculture, its utility is limited because it is less potent than ABA in binding to PYL receptors and in conferring drought stress resistance in plants[23,24].

In addition to the above-mentioned chemical methods, many researchers have investigated transgenic approaches for increasing drought resistance in plants. Overexpression of various *Arabidopsis PYL* genes and their homologs from other plants enhances drought stress resistance of transgenic plants[25,26]. However, constitutive overexpression of *PYL* genes often leads to developmental defects. For example, transgenic rice with *OsPYL5* overexpression shows reduced seed yield[27], and over-expression of *AtPYL4* and its tomato homologs results in smaller rosettes in the transgenic *Arabidopsis* seedlings[26,28]. These results indicate that high level expression of *PYL* transgenes may need to be conditional and regulated to minimize the adverse effects of the genes while increasing drought resistance.

To search for more effective ABA analogs that could provide strong drought resistance in plants, we used a rational design approach to optimize molecules based on the AM1 backbone, i.e., we considered how to best fit the molecules into the space in the PYL ligand-binding pocket in order to maximize hydrogen bonding and other interactions between the molecules and the receptor. Here, we describe a series of novel compounds, named AMFs, in which fluorine atoms are added to the benzyl ring of the AM1 backbone. These new PYL receptor agonists greatly improve the binding affinity to PYLs and are more stable and effective than ABA or AM1 in conferring drought resistance to plants. We also report that application of AMFs to transgenic *Arabidopsis* and soybean plants with abiotic stress-inducible AtPYL2 over-expression dramatically increases drought resistance. Our results show that the combining of chemical and genetic approaches is an effective way to protect plants under drought stress.

## Results

**Protein structure-based optimization of AM1.** Previous studies revealed that relative to ABA, AM1 forms fewer hydrogen bonds with residues in the PYL2 ligand-binding pocket[23,24]. ABA forms direct or water-mediated hydrogen bonds with the residues in the receptor pocket via three groups (carbonyl, hydroxyl, and carboxyl) (Fig. 1). AM1, in contrast, contains only a carbonyl and a sulfonamide that function as counterparts of ABA's carbonyl and hydroxyl groups in forming hydrogen bonds with residues in PYL receptors; AM1 lacks a potential hydrogen-bonding site at the 4-methylphenyl to mimic the carboxylic group in ABA. The PYL2–AM1–HAB1 complex has a cavern around the 4-methylbenzyl ring, which leaves room for side-chain modifications[23]. It is thus possible that the binding affinity of AM1 for PYL2 may be increased by introducing hydrogen bond

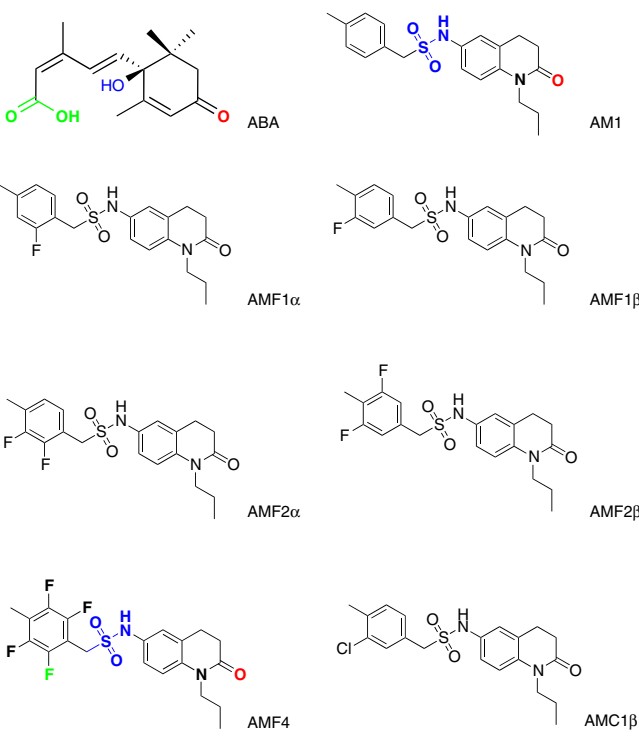

**Fig. 1** Two-dimensional chemical structure of AMFs and AMC1β. As halide derivatives of AM1, AMFs and AMC1β share a similar AM1 structural motif with one or more halogen atoms (fluorine or chloride) added in the 4-menthylphenyl tail. The three hydrogen bond-forming groups of ABA, carbonyl, hydroxyl, and carboxyl, and their counterparts in AM1 and AMF4, are highlighted in red, blue, and green, respectively

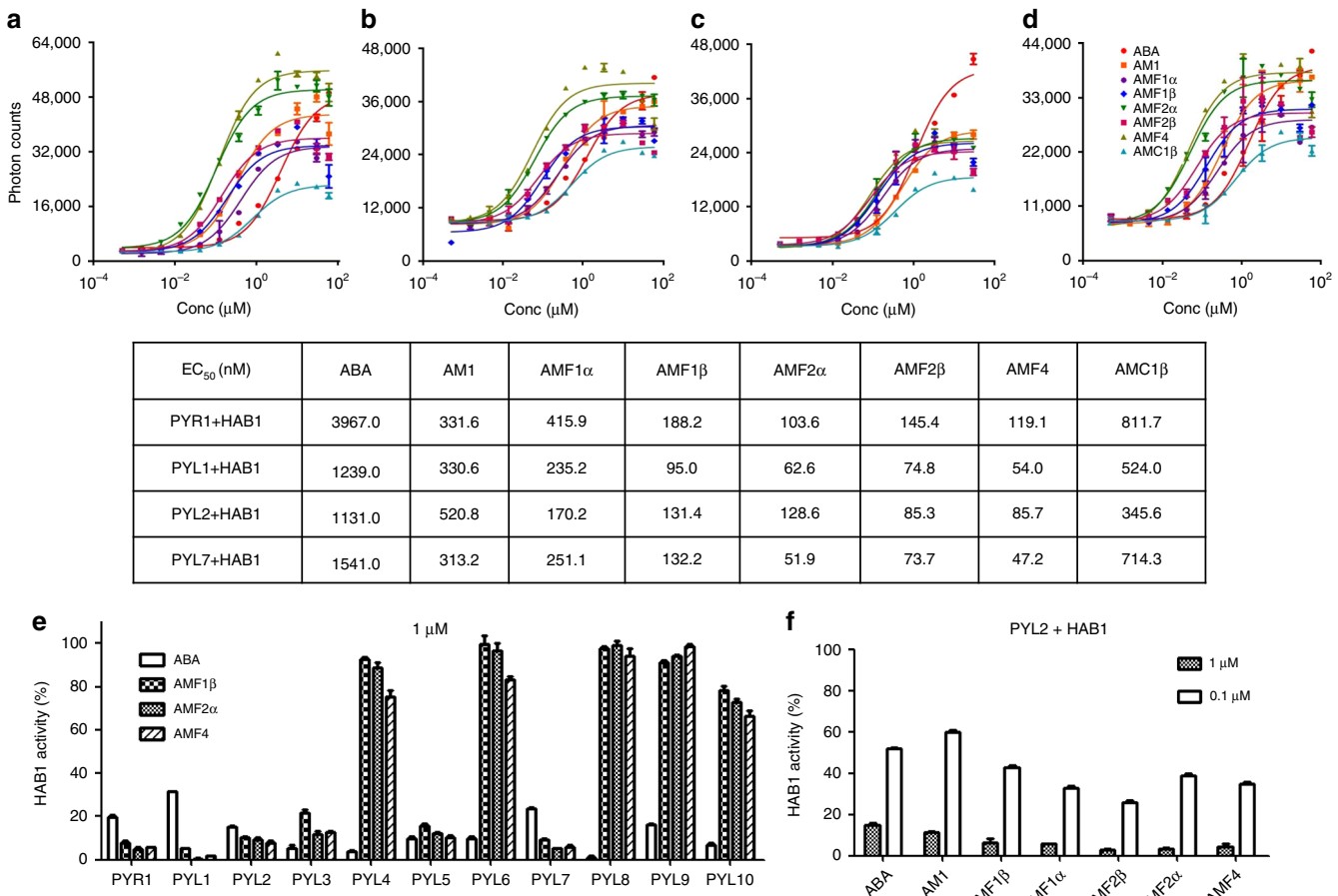

| EC$_{50}$ (nM) | ABA | AM1 | AMF1α | AMF1β | AMF2α | AMF2β | AMF4 | AMC1β |
|---|---|---|---|---|---|---|---|---|
| PYR1+HAB1 | 3967.0 | 331.6 | 415.9 | 188.2 | 103.6 | 145.4 | 119.1 | 811.7 |
| PYL1+HAB1 | 1239.0 | 330.6 | 235.2 | 95.0 | 62.6 | 74.8 | 54.0 | 524.0 |
| PYL2+HAB1 | 1131.0 | 520.8 | 170.2 | 131.4 | 128.6 | 85.3 | 85.7 | 345.6 |
| PYL7+HAB1 | 1541.0 | 313.2 | 251.1 | 132.2 | 51.9 | 73.7 | 47.2 | 714.3 |

**Fig. 2** AMFs are potent PYL receptor agonists. **a–d** Agonist dose–response curves for AMFs and AMC1β. Dose-dependent interactions between HAB1 and PYR1 (**a**), PYL1 (**b**), PYL2 (**c**), or PYL7 (**d**) in the presence of all five AMFs, AMC1β, AM1 and (+)-ABA are determined in AlphaScreen assays. EC$_{50}$ values for the interactions are listed below the curves ($n = 3$, error bars = SD). **e** Inhibition of HAB1 activity induced by AMFs and mediated by the 11 PYLs in phosphatase activity assays. The working concentration is 1 μM for AMF1β, AMF2α, and (+)-ABA. Values are means ± SD ($n = 3$). **f** Dose-dependent inhibition of HAB1 activity resulting from the binding of AMFs to the PYL2–HAB1 complex. The working concentrations are 1 and 0.1 μM for all five AMFs and (+)-ABA. Values are means ± SD ($n = 3$)

connections at the 4-methylbenzyl ring. We therefore added one, two, and four fluorine atoms in the ortho- and/or meta-positions in the 4-methylphenyl ring. The newly designed molecules were predicted to better occupy the PYL2 ligand-binding pocket, such that the added fluorine may act as a counterpart of ABA's carboxylic group in forming hydrogen bonds with surrounding residues of PYL2 (Fig. 1). These structures should still maintain the hydrogen bond-mediated interaction between the carbonyl in the quinolinone group and the Trp385 of the HAB1 protein, which is critical for inhibition of PP2C activity. We named this new series of compounds AMFs (AM1 fluorine derivatives). We characterized in detail five AMFs in our experiments, and we refer to them as AMF1α, AMF1β, AMF2α, AMF2β, and AMF4, based on the number and position of fluorine atoms (Fig. 1). A control compound with a chlorine atom in the meta-position of 4-methylbenzyl ring is referred to as AMC1β (Fig. 1).

**AMFs are potent agonists and bind PYL with high affinity.** The PYL-binding affinity of AMFs was assessed by the AlphaScreen assay, and the activity of AMFs was assessed by a PP2C (HAB1) phosphatase activity inhibition assay[16]. Four PYL proteins, PYR1, PYL1, PYL2, and PYL7, were used in this assay because they show ABA-dependent binding to HAB1 and are thus suitable for the assay. Dose–response curves and EC$_{50}$ values based on the

AlphaScreen assay revealed that the binding affinities to the PYL–HAB1 complexes were much greater for AMFs than for ABA, and mostly were also greater than for AM1 (Fig. 2a–d). Thus, the AMFs were more effective than ABA in promoting PYL–PP2C interactions. The binding affinities with the PYL receptors were largely correlated with the number of fluorine atoms, and the EC$_{50}$ values were lower for AMF2α, AMF2β, and AMF4 than for AMF1α and AMF1β. The EC$_{50}$ values also indicated that the binding affinities to the four tested PYL receptors for AMC1β were lower than for the AMFs, and were mostly also lower than for AM1, although the AMC1β binding affinities were still higher than those of ABA (Fig. 2a–d). These results suggested that chloride is less effective than fluorine in AM1 modification. We also examined the binding of some of the chemicals to two soybean PYLs, GmPYL3 and GmPYL6, which were orthologs of AtPYL1 and AtPYL2, respectively[29]. Consistent with the results obtained for *Arabidopsis* PYLs, EC$_{50}$ values for binding to GmPYL–AtHAB1 complexes were significantly lower for all the tested AMFs than for ABA (Supplementary Fig. 1). These results indicate that PYL receptors are sufficiently conserved such that the newly designed molecules may be applicable to diverse plant species.

We selected 11 of the 14 *Arabidopsis* PYLs to compare the activities of ABA and three AMFs (AMF1β, AMF2α, and AMF4) in causing inhibition of HAB1 activity (Fig. 2e). As expected,

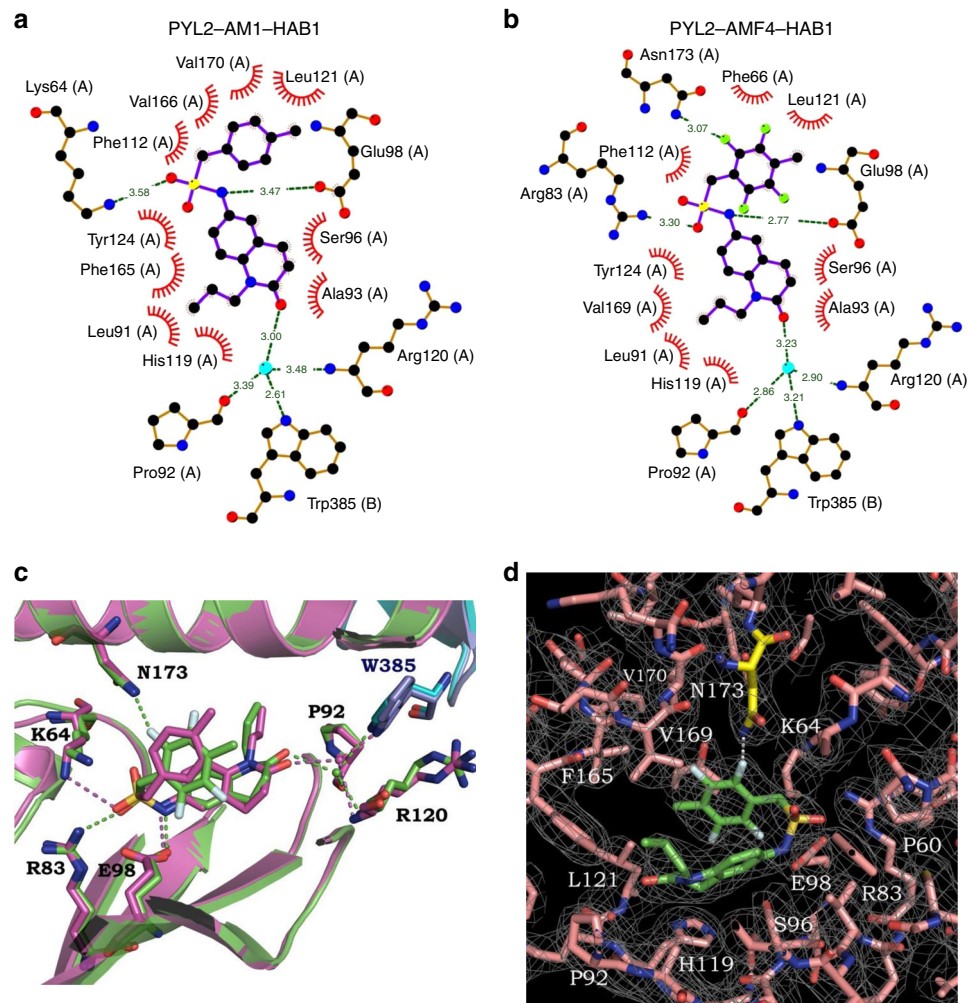

**Fig. 3** Structural comparison of AM1 and AMF4 within the PYL2–HAB1 complex. **a**, **b** Two-dimensional structural schematics of interactions between AM1 (**a**) or AMF4 (**b**) and residues in the PYL2 ligand-binding pocket (A) or in HAB1 (B). The schematics show an increase in the number of hydrogen bonds (dashed lines) between the fluorine atoms (green-filled circles) of AMF4 and the nitrogen atoms (blue-filled circles) of the Asn173 residue in the PYL2-binding pocket. Red-, yellow-, blue-, and turquoise-filled circles represent oxygen atoms, sulfur atoms, nitrogen atoms and water molecules, respectively. The number represents the distance (Å) between two atoms/molecules. The PYL2–AM1–HAB1 schematic model is from previously published paper[23]. **c** Overlay of three-dimensional structural schematics of AM1 (purple) with AMF4 (green) in the PYL2-binding pocket, with hydrogen bonds (dashed lines) and water molecules (filled circles) for AM1 and AMF4 in purple and green colors, respectively. **d** A $2F_o$–$2F_c$ electron density map of bound AMF4 and its surrounding residues contoured at $1.0\sigma$. N173 of PYL2 is highlighted in yellow and the white dotted line represents the hydrogen bond between the fluorine atom of AMF4 and N173 of PYL2

ABA bound to all 11 of the tested PYLs and inhibited HAB1 activity at 1 μM concentration. The HAB1 inhibition was greater with AMF1β and AMF2α than with ABA when bound to PYR1, PYL1, PYL2, or PYL7, but was equivalent or slightly less than with ABA when bound to PYL3 or PYL5. AMF4 caused a strong inhibition of HAB1 when bound to PYR1, PYL1, PYL2, PYL3, PYL5, or PYL7. AMF4 also caused HAB1 inhibition when bound to PYL4, PYL6, or PYL10, although the inhibition was much less than that caused by ABA. AMF1β and AMF2α also slightly inhibited HAB1 when bound to PYL10. AMF4 did not significantly inhibit HAB1 in the presence of PYL8 or PYL9, and AMF1β and AMF2α did not inhibit HAB1 in the presence of PYL4, PYL6, PYL8, or PYL9 (Fig. 2e). PYL2–HAB1 and the AMFs interacted in a dose-dependent manner, with greater AMF-mediated inhibition of HAB1 activity at 1 μM than 0.1 μM. In all cases, inhibition of HAB1 activity was greater by AMF than by ABA or AM1 at equivalent concentrations (Fig. 2f).

**Structures of PYL2–AMF–HAB1**. Analysis of the crystal structure of the PYL2–AM1–HAB1 complex revealed that AM1 bound to the ligand-binding pocket in PYL2 via hydrogen bonds, causing closure of the "gate-latch" structure[23]. The conserved tryptophan (Trp385) residue in HAB1 further secured this "gate-latch" mechanism[23] (Fig. 3a). To elucidate how AMFs stimulate PYL–PP2C interactions, we compared the crystal structures of PYL2–AMFs–HAB1 complexes (Fig. 3b and Supplementary Fig. 2b–f) with those of PYL2–AM1–HAB1[23] (Fig. 3a) and PYL2–ABA–HAB1 complexes (Supplementary Fig. 2a) at 2.3- to 2.6-Å resolution (Supplementary Table 2). PYL2–AMFs–HAB1 and PYL2–AM1–HAB1 shared a similar structural motif. The fluorous benzyl group, sulfonamide link and di-hydro quinolinone ring of AMFs fitted snugly into the hydrophobic pocket of PYL2 (Fig. 3 and Supplementary Fig. 2c–f). For AMF2α (Supplementary Fig. 2d) and AMF4 (Fig. 3b), fluorine atoms in the ortho-position of the 4-methylbenzyl ring formed an extra hydrogen bond with the Asn173 residue, which also formed

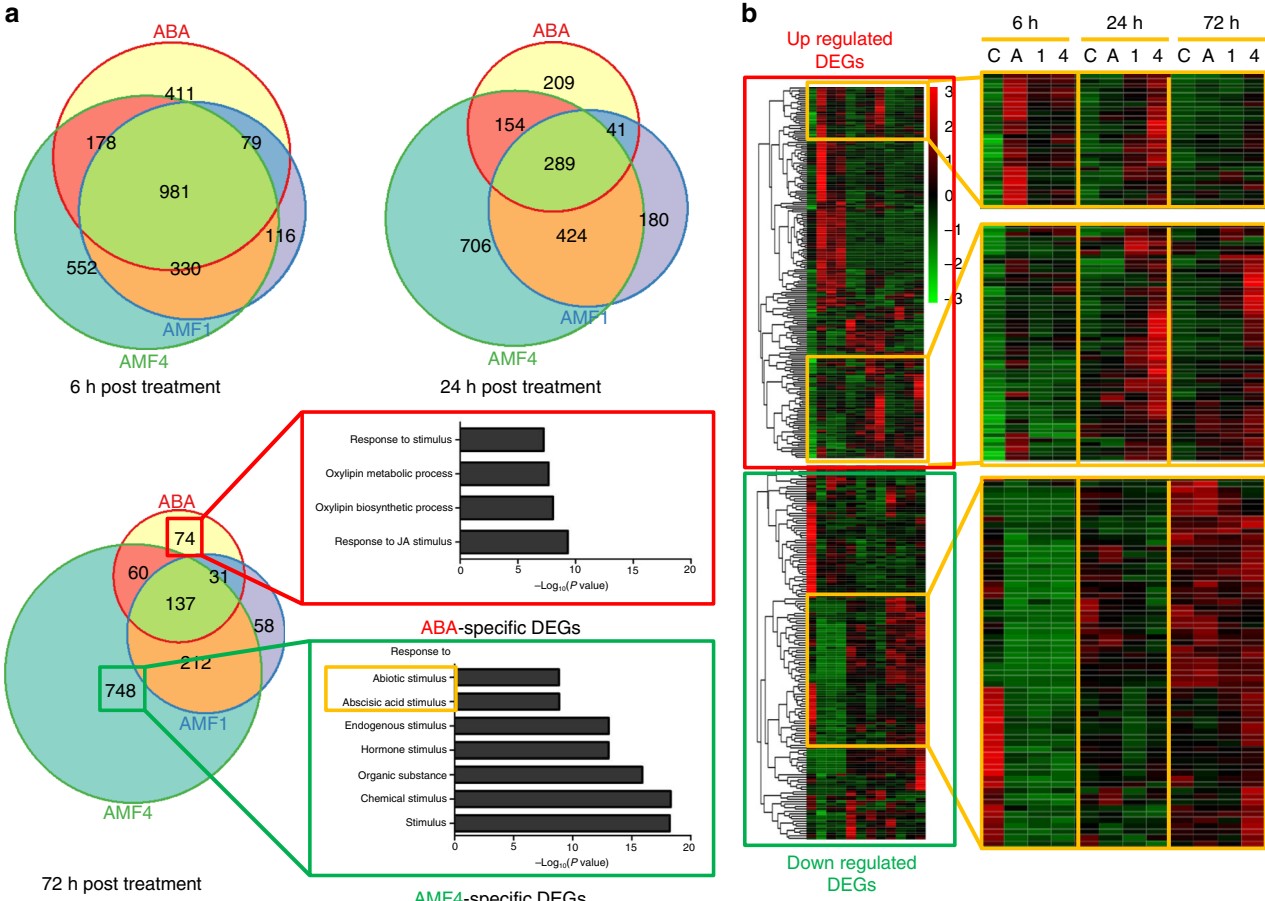

**Fig. 4** AMFs regulate ABA-responsive genes in *Arabidopsis* plants. Col-0 plants are sprayed with 10 μM (+)-ABA (ABA or A), AMF1β (AMF1 or 1), or AMF4 (AMF4 or 4) are sampled 6, 24, and 72 h post treatment, and DMSO (DMSO or C) is used as the control. Gene expression profiles are based on RNA-seq results. **a** Overlap of ABA- and AMFs-induced DEGs. AMF1β or AMF4 induce highly correlated responses at the transcript level compared with ABA. Ontology analysis of ABA- or AMF4-specific DEGs shows that some ABA- or abiotic stress-related processes are still among the most enriched biological processes (based on their *P* values) at 3 days after AMF4 treatment but not at 3 days after ABA treatment. **b** Heat map of ABA- or abiotic stress-related DEGs in Col-0 plants. DEGs are clustered based on expression profiles. Each row contains an ABA- or abiotic stress-related DEG based on gene ontology analysis. Each column represents a chemical-time combination. The DEGs showing divergent time-course response to ABA or AMFs are highlighted. The color in each cell indicates its relative expression level compared to the mean expression level in all samples of the same row according to the color bar, with red and green representing upregulated and downregulated genes, respectively

water-mediated hydrogen bonds with the carboxylic group of ABA in the PYL2–ABA–HAB1 complex (Supplementary Fig. 2a). Asn173 is critical in ABA–PYL2 interaction[16], and our data show that when Asn173 was substituted with Ala, PYL2 (N173A) and HAB1 interactions in the presence of all the AMF and AMC1β compounds were nearly abolished (Supplementary Fig. 3). The relatively weak hydrogen bonds between the sulfonamide group and the surrounding residues in the PYL2-binding pocket of the PYL2–AM1–HAB1 complex were strengthened in PYL2–AMF/AMC1β–HAB1 complexes. The lengths of hydrogen bonds between sulfonamide nitrogen in AMF/AMC1β compounds and Glu98 were reduced to 2.8–2.9 Å, compared to 3.5 Å in the PYL2–AM1–HAB1 complex. Further, the weak bond (3.6 Å) between sulfonyl oxygen and Lys64 in the PYL2–AM1–HAB1 complex was absent in PYL2–AMFs–HAB1 and PYL2–AMC1β–HAB1 complexes. Instead, new hydrogen bonds were formed between the sulfonyl oxygen and nitrogen atoms on the charged side chains of the Arg83 residue (Fig. 3b and Supplementary Fig. 2b–f), indicating that the introduced fluorines in the 4-methylbenzyl ring also caused steric hindrance to enable the compounds to fit more snugly into the hydrophobic cavern of PYL2. This steric hindrance may also weaken the water-mediated

hydrogen bond between sulfonyl oxygen and Asn173 in PYL2–AMF1β–HAB1 (Supplementary Fig. 2c). Introduction of fluorine atoms into the 4-methylbenyl group thus led to the formation of more or stronger hydrogen bonds between AMFs and the PYL2 pocket than between AM1 and the PYL2 pocket, resulting in much higher PYL2-binding affinities for the AMFs (Fig. 2c). There was a direct relationship between the number of fluorine atoms in these compounds, especially in the ortho-position of the 4-methylbenyl ring, and the number of direct hydrogen bonds formed. This relationship helped explain the positive correlation between fluorine number and PYL2–HAB1 complex-binding affinity shown in Fig. 2a–d. The water-mediated hydrogen bonds between the carbonyl oxygen of the di-hydro quinolinone ring and the Trp385 residue of HAB1 were presumably important for the closure of the "gate-latch-lock" structure in all PYL2–AMF–HAB1 complexes (Fig. 3 and Supplementary Fig. 2c–f).

**AMFs activate the expression of ABA-responsive genes.** To characterize AMF effects in *Arabidopsis*, we used RNA sequencing (RNA-seq) to profile the transcriptomes of 3-week-old plants

treated with 10 μM ABA, AMF1β, AMF4, or DMSO (control). RNA samples were isolated before treatment and 6, 24, or 72 h post treatment. Col-0 wild-type and p*RD29a-PYL2* transgenic plants with drought-inducible overexpression of *PYL2* were also used for RNA-seq. In the analysis, differentially expressed genes (DEGs) were those with fold change >2 or <0.5 compared with the DMSO control sample. The results showed that at 6 h post treatment, 1238 of the 1649 (75%) DEGs in response to ABA in Col-0 wild-type plants were also DEGs in response to AMF1β or AMF4, and that 981 of the 1649 genes (59%) of the DEGs were responsive to both AMF1β and AMF4 (Fig. 4a). Although the total number of DEGs decreased at 3 days post treatment, more than 75% of the DEGs for ABA (228 of 302 genes) were also DEGs for AMF1β or AMF4 at that time (Fig. 4a), indicating that expression profiles were correlated between ABA and AMFs ($R^2 = 0.76$ for AMF1β and 0.64 for AMF4, with a cut-off false discovery rate (FDR)<0.05) (Supplementary Fig. 4). Interestingly, the changes in expression persisted much longer after treatment with AMFs than with ABA. Gene ontology analysis revealed that ABA- and abiotic stress-related processes were still highly enriched in AMF4- but not ABA-specific DEGs at 3 days after treatment (Fig. 4a). Consistent with the gene ontology analysis, a heat map of ABA- or abiotic stress-related DEGs showed that most of the genes induced by ABA in the first 6 h decreased to basal levels 1 day after ABA or AMF treatments, but that some of the genes maintained high expression levels in plants treated with AMFs (Fig. 4b). ABA- and abiotic stress-responsive genes that remained highly induced at 3 days after AMF4 but not ABA treatment included MYB60[30], RD26[31], HAI1[32], RD28[33], RD29b[34], RAB18[35,36], LEA4-5[37], P5CS1[38], NCED3[39], SnRK3.10[40], and MPK3[41,42], which have been shown to function in abiotic stress resistance. These results indicated a longer lasting effect of AMF4 in the induction of genes involved in abiotic-stress resistance. Consistent with this longer lasting effect of AMF4, we found that AMF4 was physiologically more stable than ABA and AM1. At 12 h after treatment, ~30% AMF4 were detected in plants, whereas only ~16% AM1 and ~4% ABA remained (Supplementary Fig. 5).

**AMFs enhance plant drought resistance**. Previous reports showed that 1 μM ABA inhibited seed germination in the Col-0 wild type but less so in the *pyr1;pyl1;pyl4* triple mutant[6,23]. This observation indicated that PYR1, PYL1, and PYL4 were critical for ABA-mediated inhibition of seed germination. AM1 also inhibited seed germination[23]. Similarly, AMFs inhibited seed

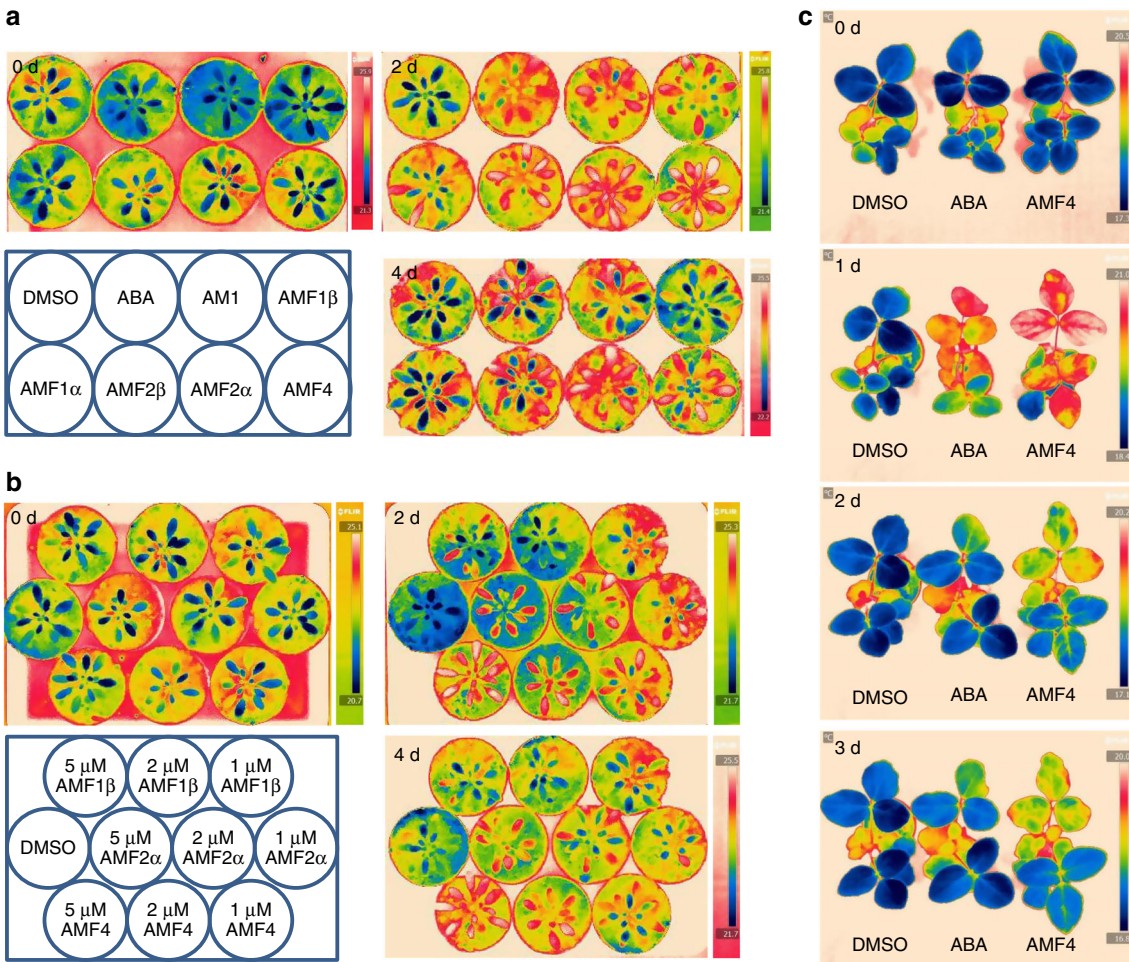

**Fig. 5** AMF sprays increase leaf temperature. **a** Leaf surface temperature is increased by treatment with AMFs. Four-week-old *Arabidopsis aba2-1* mutants are sprayed with 5 μM indicated chemical solutions, and DMSO is used as the control. Plants are photographed with an IR camera before treatment (0 days) and at 2 or 4 days after treatment. **b** AMF sprays increase leaf temperature in a dose-dependent manner. Four-week-old *Arabidopsis aba2-1* mutants are treated with corresponding chemicals at 5, 2, or 1 μM, and DMSO is used as the control. Plants are photographed with an IR camera before treatment (0 days) and at 2 or 4 days after treatment. **c** AMF4 increases soybean leaf temperature. Soybean plants are treated with 20 μM (+)-ABA or AMF4, and DMSO is used as the control. Plants are photographed with an IR camera before treatment (0 days) and at 1, 2, or 3 days after treatment

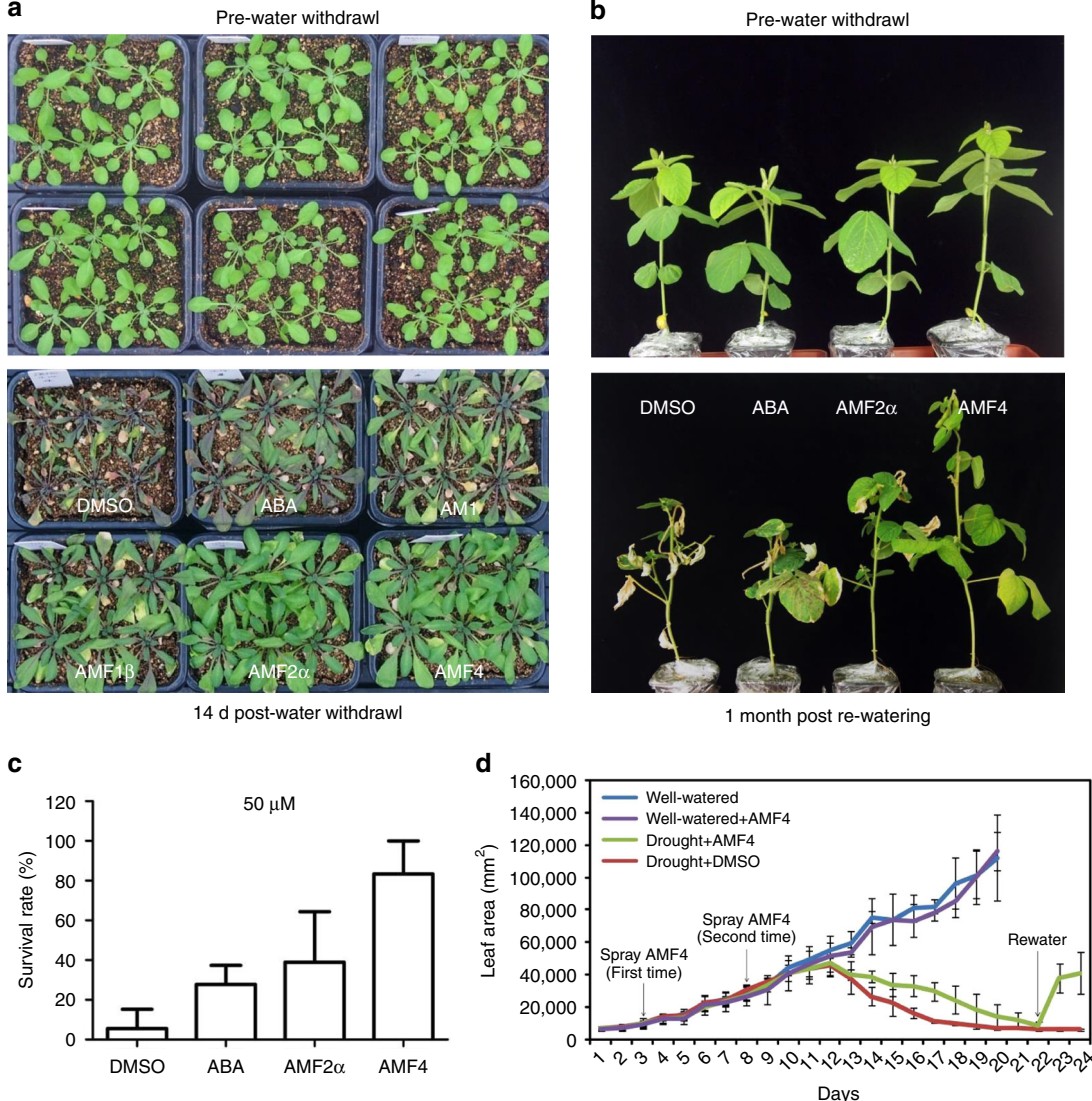

**Fig. 6** AMFs increase the drought resistance of *Arabidopsis* and soybean plants. **a** AMF treatments increase drought resistance of *Arabidopsis*. Wild-type (Col-0) plants are grown under short-day conditions for 2 weeks before watering is stopped. The plants are subsequently treated with DMSO (control), or 10 μM (+)-ABA, AM1, or AMFs once per week for another 2 weeks before watering was resumed. The plants are photographed before watering was stopped (top panel) and 14 days after watering is stopped (bottom panel). **b** AMF4 treatments increase drought resistance of soybean plants. Williams 82 soybean plants at the triple trifoliate stage are subjected to drought (watering was stopped), and then are sprayed twice at 3-days intervals with DMSO (control) or 50 μM (+)-ABA, AMF2α, or AMF4 in 1 week period before watering is resumed. The plants are photographed before watering is stopped and 1 month after watering is resumed. **c** Survival rates of plants in **b** are calculated 1 month after watering is resumed; plants are considered to have survived if they have new leaves emerging. Values are the mean survival rates from 15 individual plants per treatment, and error bars indicate SD. **d** The growth of soybean is monitored by measuring the area of all leaves. Eight-day-old Williams 82 soybean plants are subjected to drought (watering was stopped), and their leaf areas are recorded by a camera. The plants are sprayed with DMSO or AMF4 (20 μM) at 3 and 8 days after watering is stopped. For the well-watered condition, the plants are watered every 3 days. Error bars indicate the standard deviation of four biological replicates

germination in a PYR1/PYL1/PYL4-dependent manner (Supplementary Fig. 6). Germination inhibition was greater with AMF2α and AMF4 than with AMF1β or AM1, which was consistent with their receptor-binding affinities (Fig. 2a–d). Simultaneous disruption of PYR1, PYL1, and PYL4 only partially abolished germination inhibition by ABA, indicating the involvement of additional PYLs in mediating the effect of ABA on germination. Despite the much lower $EC_{50}$ values for the PYL–AMF–HAB1 interaction than for the PYL–ABA–HAB1 interaction, seed germination inhibition was greater for ABA than for AM1 and AMFs (Supplementary Fig. 6), suggesting that some PYLs that bound weakly or did not bind to AMFs contributed to ABA inhibition of seed germination. The *pyr1;pyl1;pyl4* triple mutant

seeds appeared more resistant to the AMFs than to ABA (Supplementary Fig. 6), indicating that PYL4 may not be as important as PYR1 and PYL1 for inhibition of seed germination since the AMFs were not effective on stimulating PYL4 activity (Fig. 2e).

Stomatal responses and transpirational water loss can be indirectly monitored by measuring leaf surface temperature using infrared thermal imaging[43]. To minimize the effect of endogenous ABA, we used the ABA biosynthesis-defective mutant *aba2-1* for this assay[44]. Leaf temperature showed no obvious difference for all plants before treatment (Supplementary Fig. 7a–d). At 2 days after they were sprayed on plants, ABA, AM1, and the AMFs at 5 μM all increased leaf temperature compared with the DMSO control (Fig. 5a and Supplementary Fig. 7b), indicating

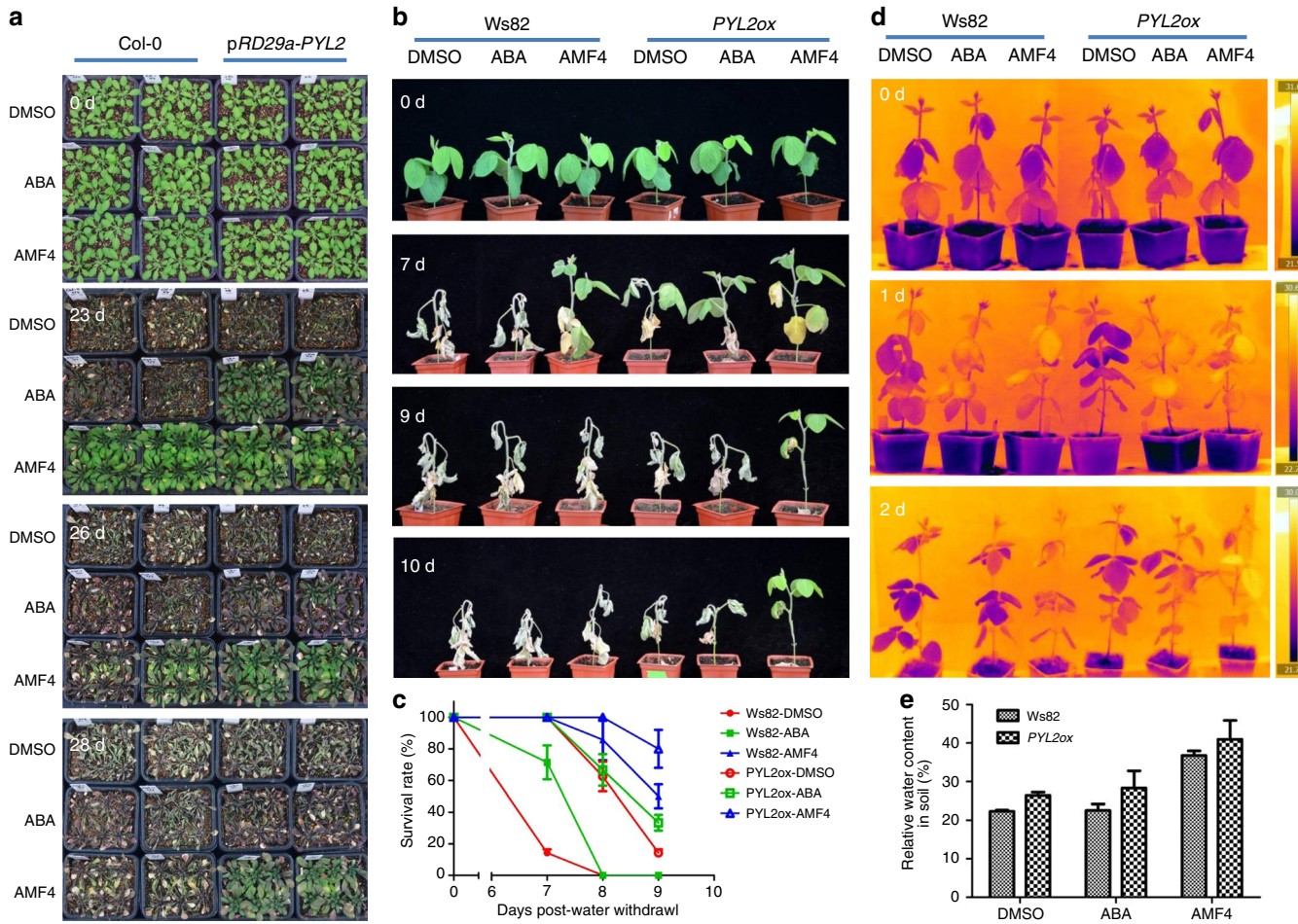

**Fig. 7** The combining of chemical and genetic approaches provides the highest drought resistance. **a** Additive effects of AMF4 and stress-induced *PYL2* expression in enhancing drought resistance in *Arabidopsis*. Wild-type plants (Col-0) and transgenic plants transformed by *Arabidopsis RD29a* promoter-driven PYL2 (p*RD29a-PYL2*) are grown under short-day conditions for 2 weeks before watering is stopped. The plants are subsequently treated with DMSO (control) or 10 μM (+)-ABA or AMF4 once per week for another 2 weeks. The plants are photographed before treatment (0d) and at 23, 26 or 28 days after first chemical treatment. **b–e** Additive effects of AMF4 and stress-induced *PYL2* expression in enhancing drought resistance in soybean. **b** Wild-type (Ws82) and transgenic plants transformed by *RD29a* promoter-driven *PYL2* (*PYL2ox*) at the triple trifoliate stage are subjected to drought (watering was stopped) and are treated every 3 days with DMSO (control), 50 μM (+)-ABA, or 50 μM AMF4 for 9 days before watering is resumed. The plants are photographed before watering is stopped (0 days) and 7, 9, and 10 days after watering is stopped. **c** Survival rates are calculated daily from the 7th day until the 9th day after watering is stopped, and plants are considered to have survived if their leaves had not wilted. Values are the mean survival rates from three replicates with 15 individual plants per treatment, and error bars indicate SD. **d** Soybean plants treated with 20 μM (+)-ABA, AMF4, or DMSO (the control) are photographed with an IR camera before they are treated (0 days) and 1 or 2 days after treatment. **e** The relative soil water content in **b** is determined as the mass of water in soil/mass of oven-dry soil*100%, and values are the mean soil water content after 7 days without watering, when DMSO-treated wild-type soybean plants begin to wilt. Each combination of treatment and plant line is represented by six replicate pots, and error bars indicate SD

that these compounds increased stomatal closure. Importantly, the effects of AMF2α and AMF4 persisted for 2 days longer than those of the other compounds (Fig. 5a and Supplementary Fig. 7c). We then compared the effects of spraying plants with AMF4, AMF2α, and AMF1β at 1, 2, and 5 μM. At 2 days post treatment, all plants treated with these chemicals showed an elevated temperature compared to the DMSO control (Fig. 5b and Supplementary Fig. 7e). At 4 days post treatment, plants treated with 2 and 5 μM AMF4 or 5 μM AMF2α still exhibited an elevated leaf temperature (Fig. 5b and Supplementary Fig. 7f). Together, these results showed that AMFs had a more lasting effect than ABA or AM1 in closing stomata and in reducing transpirational water loss, and that their effect was generally correlated with fluorine atom number. For soybean plants, 20 μM AMF4 also elevated leaf temperature and was more effective than ABA; as with *Arabidopsis*, the effect with soybean persisted longer

for AMF4 than ABA (Fig. 5c and Supplementary Fig. 7g). In addition, the older soybean leaves appeared to be more sensitive to ABA and AMF4 than the younger leaves (Fig. 5c).

Consistent with their effects on ABA-responsive genes and leaf transpiration, treatments with AMFs also increased drought resistance in plants. Drought resistance of *Arabidopsis* Col-0 plants was greater following treatment with 10 μM AMF1β, AMF2α, or AMF4 than treatment with DMSO, ABA, or AM1 (Fig. 6a and Supplementary Fig. 8a). AMF4-treated plants were more drought resistant than AMF2α/2β-treated plants (Supplementary Fig. 8b) or AMF1β-treated plants (Fig. 6a). Soybean plants treated with 50 μM AMF4, AMF2α, or ABA were also more resistant to drought than plants treated with DMSO, and survival after drought stress was greater for AMF4-treated plants than for ABA- or AMF2α-treated plants (Fig. 6b, c). The leaf area of soybean plants was assessed after the plants were sprayed with

20 μM AMF4 or the DMSO control and were then subjected or not subjected to drought stress. Without drought stress, the AMF4 sprays did not affect soybean leaf growth (Fig. 6d). With drought stress, the soybean plants sprayed with AMF4 or DMSO control began to wilt at day 13 as indicated by decreases in their leaf areas (Fig. 6d). However, the decrease in leaf area was slower for the AMF4-treated plants, and the AMF4-treated plants but not the DMSO-treated plants recovered after rewatering on day 22 (Fig. 6d).

**PYL2 overexpression enhances the effect of AMFs.** PYL2 is very effective in causing PP2C inhibition when bound to the AMFs (Fig. 2e and f). We generated *Arabidopsis* and soybean transgenic lines with *AtPYL2* overexpression driven by the ABA- and drought-responsive *RD29a* promoter[34] (p*RD29a-PYL2*) (Supplementary Fig. 9). The *Arabidopsis* p*RD29a-PYL2* transformants showed greater resistance to drought stress than the Col-0 wild type, and ABA or AMF treatment further enhanced the drought resistance phenotype (Fig. 7a and Supplementary Fig. 10). At 23 days after water withdrawal, all DMSO-treated Col-0 plants, DMSO-treated p*RD29a-PYL2* transformants, and 10 μM ABA-treated Col-0 appeared dead, whereas 10 μM ABA-treated p*RD29a-PYL2* transformants were still alive, and Col-0 and p*RD29a-PYL2* transformants treated with 10 μM AMF4 appeared to be in the best condition (Fig. 7a). AMF4-treated p*RD29a-PYL2* transformants were alive even at 28 days after water withdrawal, when all other plants had died (Fig. 7a). Similarly, p*RD29a-PYL2* transformants treated with 20 μM AMF1β also survived drought stress much longer than Col-0 plants treated with 20 μM AMF1β or ABA, or p*RD29a-PYL2* transformants treated with 20 μM ABA (Supplementary Fig. 10a). At 18 days after water withdrawal, the soil water content (g of water per 100 g of dry soil) was >20% for the AMF4-treated Col-0 plants but was <10% for ABA- and DMSO-treated Col-0 plants (Supplementary Fig. 10b). The soil water content was >25% for AMF4-treated p*RD29a-PYL2* transgenic plants and was >20% for ABA-treated p*RD29a-PYL2* transgenic plants, but was <6% for DMSO-treated p*RD29a-PYL2* transgenic plants, at 18 days after water withdrawal (Supplementary Fig. 10b). These results were consistent with the ABA- and AMF4-induced transpiration decreases in the treated plant leaves.

p*RD29a-AtPYL2* transgenic soybean plants were more drought resistant than the wild-type Williams 82 plants (Fig. 7b). One week after water withdrawal, most wild-type soybean plants treated with DMSO and some treated with 50 μM ABA had wilted (Fig. 7c). DMSO-treated p*RD29a-PYL2* transgenic soybean plants also began to wilt. However, wild-type soybean plants treated with 50 μM AMF4 or p*RD29a-PYL2* transgenic soybean plants treated with ABA or AMF4 remained unwilted and grew well (Fig. 7b, c). Nine days after water withdrawal, about 50% of the wild type and >80% of the p*RD29a-PYL2* transformants treated with 50 μM AMF4 survived, but all of the wild-type plants and most of the p*RD29a-PYL2* transgenic plants treated with DMSO or ABA had died (Fig. 7b, c). The strong drought resistance phenotype of the AMF4-treated p*RD29a-PYL2* transformants was at least partly due to reduced transpirational water loss. Although both transgenic and wild-type soybean plants had increased leaf temperatures relative to the control 1 day post-ABA or -AMF4 treatment, AMF4 had a stronger effect on the p*RD29a-PYL2* plants than on the wild-type plants. Two days post treatment, leaf temperature was highest for AMF4-treated p*RD29a-PYL2* plants (Fig. 7d). The relative water content in soil also reflected the changes in transpiration. Seven days after water withdrawal, water content was highest in the soil with AMF4-treated p*RD29a-PYL2* transgenic plants and was lowest in the soil

with DMSO-treated wild-type soybean plants (Fig. 7e). These results demonstrated that treatment with the AMFs combined with stress-inducible overexpression of *PYL2* provided a very high level of protection for plants under drought stress.

**Discussion**

In this study, we designed ABA agonists with the goal of creating compounds that more effectively increase the drought resistance of plants than the previously reported ABA agonist, AM1[23,24]. By adding fluorine atoms to the 4-methylbenyl ring of AM1, we created compounds, named AMFs, that fitted more snugly than AM1 into the ligand-binding pocket of the ABA-receptor PYL proteins and that significantly increased the ligand–receptor binding affinity due to formation of extra hydrogen bonds. Our newly designed AMFs dramatically increased drought resistance in both *Arabidopsis* and soybean when compared with ABA and AM1. A combined chemical and genetic approach using AMF treatment on plants with inducible *PYL* overexpression proved to be an even more effective way for enhancing drought resistance.

Based on the structure of PYL2–AM1–HAB1[23], we reasoned that introducing fluorines to the 4-methylbenzyl ring of AM1 may improve the ligand-binding affinity for PYL–PP2C co-receptors. Indeed, the binding affinities of the fluorine-added compounds to the four tested *Arabidopsis* PYL receptors (PYR1, PYL1, PYL2, and PYL7) were generally one order of magnitude higher than those of ABA, and the binding affinities with the PYL receptors were largely correlated with the number of fluorine atoms. These results and results from the HAB1 activity inhibition assays indicated that the newly designed AMFs were more effective agonists of PYL receptors than AM1. AMFs also had greater binding affinities than ABA to soybean PYL receptors, which was consistent with the highly conserved nature of PYLs in higher plants[18].

The HAB1 inhibition assays suggested that the AMFs bound with high affinities to PYR1 and PYLs 1, 2, 3, 5, and 7 but not PYLs 4, 6, 8, 9, and 10. The selectivity of AMFs for the PYLs implies that PYR1 and PYLs 1, 2, 3, 5, and 7 share certain structural features that are different in PYLs 4, 6, 8, 9, and 10. The PYL selectivity of AMFs may explain why the AMFs were less effective than ABA in seed germination inhibition and why the *pyr1;pyl1;pyl4* triple mutant is more resistant to the AMFs than to ABA. Besides PYR1 and PYL1, some other PYLs that do not bind AMFs are also important for ABA inhibition of seed germination.

The crystal structures of PYL2–AMF–HAB1 complexes indicated why the addition of halogen atoms increased the binding affinities of AMFs to PYL–HAB1 co-receptors. As shown in the PYL2–AMF2α–HAB1 and PYL2–AMF4–HAB1 complexes, a direct hydrogen bond formed between the fluorine in the ortho-position of the 4-methylbenzyl ring and the surrounding Asn173 residue. With this additional hydrogen bond, AMF2α and AMF4 contained counterparts of all three potential hydrogen bonding sites in ABA: carbonyl, hydroxyl, and carboxylic. Introducing halogens into the 4-methylbenzyl ring resulted in steric hindrance and therefore also altered existing hydrogen bonds in the PYL2–AM1–HAB1 complex, such that the weak hydrogen bond between sulfonyl oxygen and Lys64 was replaced by a stronger bond between sulfonyl oxygen and Arg83, and the hydrogen bond length between the sulfonamide nitrogen and the Glu98 residue was reduced compared to that in the PYL2–AM1–HAB1 complex. These structural changes explained well the increased binding affinities of AMFs and AMC1β compounds to the PYL–HAB1 co-receptors. In addition, the fluorine in the meta-position of the 4-methylbenzyl group also provided spatial support so that the other fluorine in ortho-position located to a proper distance and angle to approach Asn173, which was the

case for AMF2α and AMF4. The lack of this spatial support can explain the missing hydrogen bond between the fluorine in the ortho-position and Asn173 in the PYL2–AMF1α–HAB1 complex. The direct hydrogen bonds between AMF/AMC1β compounds and surrounding residues of the PYL2 pocket were stronger but less flexible than the water-mediated hydrogen bonds in the PYL2–ABA–HAB1 complex, which is consistent with the highly variable binding affinities of AMF/AMC1β compounds to different PYL proteins.

Among the AMFs tested, AMF2α and AMF4 had the highest binding affinities for most of the PYL–HAB1 co-receptors in the AlphaScreen assay. These results complemented our finding that AMF2α and AMF4 were the most effective compounds in in vivo assays, i.e., the chemicals were the most effective at reducing transpiration (as indicated by infrared imaging of leaf temperature) and at increasing drought resistance in *Arabidopsis*. AMF4 was also more effective than ABA in reducing transpiration and increasing drought resistance in soybean.

This is the first report of the creation of artificial PYL ligands that are much superior to the natural one, ABA, in protecting plants from drought stress. In addition to mediating hydrogen bond formation, the fluorine atoms may also affect compound liposolubility, and the chemicals with more fluorine atoms could be more permeable across the plasma membrane[45]. More hydrogen bonds with residues in the PYL-binding pocket and potentially increased liposolubility may explain why AMF2α and AMF4 were more effective than the other AMFs in the in vivo assays.

RNA-seq results showed that gene expression profiles induced by AMF1β and AMF4 were highly correlated with that induced by ABA. Both gene ontology and heat map analyses showed that many ABA-inducible genes maintained high expression levels for much longer time in AMF4-treated plants. Our data further showed that AMF4 was more stable in plants than AM1 and ABA. The C–F bond is about 17% stronger than the C–H bond (484.9 vs. 414.5 kJ mol$^{-1}$), which may render fluorine-containing compounds more chemically stable and perhaps more resistant to physiological oxidation in the cellular environment. It is likely that higher binding affinities for dedicated PYL receptors and increased stability together make AMFs superior to ABA in conferring drought resistance to plants. AMF-treated plants showed less transpirational water loss and were more resistant to drought than ABA-treated ones. The more persistent induction of drought-responsive genes in plants treated with AMFs would also contribute to increased drought resistance by increasing tolerance to cellular dehydration. Importantly, our results also showed that AMF4 treatment did not negatively affect plant growth under well-watered conditions.

Under conditions of environmental stress, plants rely heavily on PYLs to inhibit PP2C activity in order to activate ABA and stress-response pathways. Plants, however, have also evolved a negative feedback loop at the transcriptional level by downregulating PYL expression and upregulating PP2Cs expression to counter the prominent increase in PYL–PP2C interactions in the presence of ABA[46]. Like ABA, AMF treatment also downregulated PYL2 gene expression, which indicated that the effectiveness of these compounds may be further strengthened by increasing *PYL* expression levels under abiotic stress conditions. Thus, we managed to fine-tune the feedback regulation in plants by introducing the *AtPYL2* transgene under the control of the abiotic stress-inducible *RD29a* promoter. We chose PYL2 because pyrabactin, a reported ABA analog that induces an ABA-related phenotype only at the seed germination stage and not at the seedling or adult plant stage, was an agonist of PYR1 and PYL1 but an antagonist of PYL2[6,16], which indicated that PYL2 was important in ABA-mediated stress resistance in seedlings and

adult plants. Based on these observations, we created *AtRD29a* promoter-driven AtPYL2 overexpression lines in *Arabidopsis* and soybean. p*RD29a-PYL2* transformants showed neither growth defect nor other obvious developmental phenotypes compared with wild-type plants under normal growth conditions. Under ABA or AMF treatment, *AtPYL2* expression was upregulated, especially by AMF4, in the transgenic lines because of the *AtRD29a* promoter, while the native PYL genes were still downregulated. These results suggest that the transgenic plants may have enhanced response to ABA or exogenous agonist chemicals. In support of this possibility, transgenic lines showed greater drought resistance than wild-type plants, and this resistance was further enhanced by ABA or AMF treatments. The inhibition of leaf transpiration by AMF4 also lasted much longer in the p*RD29a-PYL2* transgenic soybean plants than in Ws82 wild-type soybean plants. These lasting responses would further enhance drought resistance in AMF4-treated plants. Our leaf transpiration assays and drought stress assays using transgenic and wild-type lines of *Arabidopsis* and soybean clearly showed that AMFs combined with stress-inducible *PYL2* gene overexpression greatly increased plant resistance to drought stress. Overall, our results demonstrate that the rational design of artificial ABA analogs with high PYL-binding affinities in combination with abiotic stress-inducible overexpression of PYLs will be useful for combating drought stress in plants.

## Methods

**Chemicals**. All AMF and AMC1β compounds were synthesized in our laboratory. Details are provided in Supplementary Note 1 and NMR and high-resolution mass spectrometry (HRMS) data are listed below.

AMF1α: $^1$H NMR (400 MHz, CDCl$_3$): δ 7.33 (d, $J$ = 6.8 Hz, 1H), 7.09–6.92 (m, 5H), 4.35 (s, 2H), 3.88 (m, 2H), 2.86 (t, $J$ = 6.4 Hz, 2H), 2.64 (m, 2H), 2.33 (s, 3H), 1.66 (m, 2H), 0.97 (t, $J$ = 7.2 Hz, 3H) ppm; $^{13}$C NMR (101 MHz, CDCl$_3$): δ 169.9, 159.4 (d, $J$ = 247 Hz), 139.4 (d, $J$ = 8 Hz), 137.0, 132.3 (d, $J$ = 3.1), 131.2, 129.6, 127.9, 120.8, 119.9, 118.0 (d, $J$ = 23.1 Hz), 115.8, 115.0 (d, $J$ = 19.3 Hz), 50.5, 43.7, 31.6, 25.6, 20.4, 20.2, 11.2 ppm. HRMS (m/z): [M]+ calcd. for C$_{20}$H$_{23}$FN$_2$O$_3$S, 390.1413; found, 390.1424.

AMF1β: $^1$H NMR (400 MHz, CDCl$_3$): δ 7.18 (t, $J$ = 8.4 Hz, 1H), 7.06–6.94 (m, 4H), 6.75 (s, 1H), 4.29 (s, 2H), 3.89 (m, 2H), 2.88 (m, 2H), 2.65 (m, 2H), 2.27 (d, $J$ = 1.2 Hz, 3H), 1.67 (m, 2H), 0.99 (t, $J$ = 6.4 Hz, 3H); $^{13}$C NMR (101 MHz, CDCl$_3$): δ = 169.8, 161.5 (d, $J$ = 245 Hz), 137.1, 131.7 (d, $J$ = 5.4 Hz), 131.3, 128.1, 127.7 ($J$ = 8.0 Hz), 126.3 (d, $J$ = 3.3 Hz), 125.9 (d, $J$ = 16.9 Hz), 120.9, 120.0, 117.3 (d, $J$ = 23.2 Hz), 115.7, 56.9, 43.7, 31.6, 25.6, 20.4, 14.4, 11.2 ppm. HRMS (m/z): [M]+ calcd. for C$_{20}$H$_{23}$FN$_2$O$_3$S, 390.1413; found, 390.1419.

AMF2α: $^1$H NMR (400 MHz, CDCl$_3$): δ 7.09–6.92 (m, 5H), 6.79 (s, 1H), 4.42 (s, 2H), 3.89 (t, $J$ = 7.6 Hz, 2H), 2.87 (t, $J$ = 6.8 Hz, 2H), 2.65 (m, 2H), 2.29 (s, 3H), 1.68 (m, 2H), 0.98 (t, $J$ = 7.6 Hz, 3H) ppm; $^{13}$C NMR (101 MHz, CDCl$_3$): δ 169.9, 137.2, 131.2, 128.5, 128.4, 127.2, 126.2 (br), 125.9 (br), 120.9, 120.0, 115.6, 115.5, 115.4, 50.8, 43.7, 31.6, 25.5, 20.4, 14.3, 11.2 ppm. HRMS (m/z): [M]+ calcd. for C$_{20}$H$_{22}$F$_2$N$_2$O$_3$S, 408.1319; found, 408.1324.

AMF2β: $^1$H NMR (400 MHz, DMSO-d6): δ 9.75 (s, 1H), 7.09–6.96 (m, 5H), 4.49 (s, 2H), 3.81 (t, $J$ = 7.2 Hz, 2H), 2.80 (t, $J$ = 6.8 Hz, 2H), 2.53 (t, in DMSO peak, 2H), 2.13 (s, 3H), 1.54 (m, 2H), 0.88 (t, $J$ = 7.2 Hz, 3H) ppm; $^{13}$C NMR (101 MHz, DMSO-d6): δ 169.3, 161.0 (d, $J$ = 242 Hz), 136.0, 132.9, 130.3 (t, $J$ = 10.4 Hz), 127.7, 120.2, 119.5, 116.1, 114.1 (d, $J$ = 27 Hz), 113.0 (t, $J$ = 21.3 Hz), 56.6, 42.9, 31.7, 25.4, 20.4, 11.5, 7.2 ppm. HRMS (m/z): [M]+ calcd. for C$_{20}$H$_{22}$F$_2$N$_2$O$_3$S, 408.1319; found, 408.1315.

AMF4: $^1$H NMR (400 MHz, DMSO-d6): δ 10.07 (s, 1H), 7.08–7.05 (m, 3H), 4.58 (s, 2H), 3.81 (t, $J$ = 7.2 Hz, 2H), 2.80 (t, $J$ = 6.8 Hz, 2H), 2.52 (t, in DMSO peak, 2H), 2.25 (s, 3H), 1.52 (m, 2H), 0.88 (t, $J$ = 7.6 Hz, 3H) ppm; $^{13}$C NMR (101 MHz, CDCl$_3$) δ 170.1, 146.1 (m), 143.7 (m), 137.0, 131.4, 127.9, 120.4, 119.5, 117.8 (t, $J$ = 19.0 Hz), 115.7, 105.6 (t, $J$ = 17.1 Hz), 45.9, 43.8, 31.6, 25.5, 20.4, 11.2, 7.7 ppm. HRMS (m/z): [M]+ calcd. for C$_{20}$H$_{20}$F$_4$N$_2$O$_3$S, 444.1131; found, 444.1138.

AMC1β: $^1$H NMR (400 MHz, CDCl$_3$): δ 7.39–7.32 (m, 1H), 7.25–6.88 (m, 5H), 4.28 (m, 2H), 3.89 (m, 2H), 2.86 (m, 2H), 2.66 (m, 2H), 2.32 (s, 3H), 1.67 (m, 2H), 0.99 (t, $J$ = 7.2 Hz, 3H) ppm; $^{13}$C NMR (101 MHz, CDCl$_3$): δ 169.9, 137.0, 136.7, 135.3, 134.6, 133.4, 131.3, 129.5, 128.1, 126.9, 120.7, 119.8, 115.7, 57.0, 43.8, 31.6, 25.6, 20.7, 20.0, 11.3 ppm. HRMS (m/z): [M]+ calcd. for C$_{20}$H$_{23}$ClN$_2$O$_3$S, 406.1119; found, 406.1128.

AM1 and (+)-ABA were obtained from Life Chemicals Inc. and A. G. Scientific, respectively. All compounds were dissolved in DMSO and stored at 100 mM.

**Protein preparation**. PYR1 (residues 9–182), PYL1 (residues 36–211), and PYL2 (residues 14–188), together with full-length PYL3–PYL10 from *Arabidopsis* and

with full-length GmPYL3 and GmPYL6 from soybean, were expressed in *Escherichia coli* BL21 (DE3) as recombinant fusion proteins with an H6-SUMO tag. And Biotin-labeled PP2C protein HAB1 (residues 172–511) was prepared as a recombinant fusion protein with a Biotin-MBP tag as previously described[16]. All proteins were purified following previous protocol[16].

**AlphaScreen assay**. Interactions between PYR1/PYL1/PYL2/PYL7/GmPYL3/GmPYL6 and HAB1 (PP2C) were assessed by luminescence-based AlphaScreen technology (Perkin Elmer) as previously described[16]. All reactions contained 100 nM recombinant H6-SUMO-PYL proteins bound to nickel-acceptor beads and 100 nM recombinant biotin-MBP-PP2C bound to streptavidin acceptor beads in the presence or absence of the indicated amounts of (+)-ABA, AMFs, or AMC1β compounds. For dose–response assays, the concentration of (+)-ABA and other compounds ranged from 0.5 nM to 100 μM.

**Crystallization and structure determination**. To prepare the PYL2–ligand–PP2C ternary complex, we added ligand and purified PYL2 to purified HAB1 at a 5:1:1 molar ratio in the presence of 5 mM $MgCl_2$. PYL2–ABA/AM1/AMF/AMC1β–HAB1 complex crystals were grown at room temperature (20 °C) in hanging drops containing 1.0 μL of the purified PYL2 protein and 1.0 μL of well solution containing one of the following: 0.1 M succinic acid and 15% PEG 3350; 0.2 M Di-sodium malonate and 20% PEG 3350; 0.2 M tri-sodium citrate and 20% PEG 3350; or 0.2 M magnesium formate and 20% PEG 3350. All crystals appeared within 1 day and grew to a dimension of 100–120 μm within 3–4 days. Crystals were flash frozen in liquid nitrogen. All diffraction data were collected at 100 K using an X-ray beam at BL17U beamline at the Shanghai Synchrotron Radiation Facilities[47]. The observed reflections were reduced, merged, and scaled with DENZO and SCALEPACK in the HKL2000 package. Molecular replacement was performed using the Collaborative Computational Project 4 program Phaser; Programs O and Coot were used to manually fit the protein model. Model refinement was performed with CNS and the CCP4 program Refmac5. The two-dimensional schematic was created with LigPlot software from the EMBL-EBL homepage. The statistics of data collection and the model refinement are summarized in Supplementary Table 2.

**Site-directed mutagenesis**. Site-directed mutagenesis was carried out following the QuickChange method (Stratagene). PYL2 (residues 14–188) was mutated to PYL2 (residues 14–188, N173A) using PCR with the primer pair ASNTA FP and RP whose sequences are shown in Supplementary Table 1 and the mutation was confirmed by sequencing.

**Plants materials and growth conditions**. *A. thaliana* ecotype Columbia-0 (Col-0) and the PYL triple mutant (*pyr1;pyl1;pyl4*, Col-0 background) used in germination assays were grown on half-strength MS (Murashige and Skoog) solid medium containing 1% sucrose in an environment-controlled chamber at 22 °C with a photosynthetically active radiation of 75 μmol m$^{-2}$ s$^{-1}$ and a 16-h light/8-h dark photoperiod. Col-0, the ABA-deficient mutant *aba2-1*, and the p*RD29a-PYL2* transgenic line (Col-0 background) used in gene expression analysis, transpiration assays, and drought stress resistance assays were grown in soil with an 8-h light/16-h dark photoperiod.

Soybean ecotype Williams 82 (Ws82) and the p*RD29a-PYL2* transgenic line (Ws82 background) used in gene expression analysis, transpiration assays, and drought resistance assays were grown in soil at 26 °C with a 16-h light/8-h dark photoperiod.

**HAB1 phosphatase activity assay**. In total, 100 nM Biotin-MBP-HAB1 and 500 nM corresponding H6-SUMO-tagged PYLs were pre-incubated in 50 mM imidazole, pH 7.2, 5 mM $MgCl_2$, 0.1% β-mercaptoethanol, and 0.5 μg ml$^{-1}$ BSA for 30 min at room temperature, as described before[16]. An 11-amino acid phosphopeptide (HSQPKpSTVGTP), belonging to amino acids 170–180 of SnRK2.6 with Ser175 phosphorylated, was used as the substrate of HAB1 phosphatase. Reactions were started by adding 100 μM phosphopeptide, and the quantity of phosphate released from the phosphopeptide was determined by colorimetric assay (BioVision) 35 min later.

**Gene expression analysis**. Three-week-old Col-0 *Arabidopsis* plants were sprayed with a 10 μM solution of the indicated compound and incubated for 6, 24, or 72 h before RNA extraction. Ws82 wild-type and p*RD29a-PYL2* transgenic soybean plants at the triple trifoliate stage were sprayed with 50 μM AMF4 or (+)-ABA and incubated for 6 h before RNA extraction. A 0.05% solution of DMSO was used as the control for all chemicals. Total RNAs were extracted using the TRIzol (Invitrogen) method, and RNase-free Dnase (Qiagen) was used to remove contaminating DNA before quantitative real-time PCR (qRT-PCR) and RNA-seq.

For qRT-PCR, total RNAs were reverse transcribed with the TransScript RT kit (Invitrogen) according to the manufacturer's instructions. All qRT-PCR assays were performed following the two-step protocol of the ChamQ SYBR qPCR Master Mix (Vazyme) in a CFX96 Real-time system (BIO-RAD) according to the

manufacturer's instructions (95 C*15 s, 60 C*30 s, 40 cycles). Each assay included three biological replicates and was performed twice. *AtACT7* and *GmACT2* were used as internal controls for *Arabidopsis* and soybean, respectively, in the qRT-PCR. The primer pairs used in qRT-PCR, whose sequences are shown in Supplementary Table 1, were AtPYL2 qF and qR, AtACT7 qF and qR, and GmACT2 qF and qR for *AtPYL2*, *AtACT7* and *GmACT2*, respectively, with a final concentration of 0.2 μM.

For RNA-seq, duplicate biological replicate samples were used for DMSO, AMF, and ABA treatment. Total RNA samples were sequenced by the genomics core facility of the Shanghai Center for Plant Stress Biology using the HiSeq2500 system (Illumina), with 10 M reads per sample with average length >49 bp. All reads were mapped to the *Arabidopsis* genome (TAIR10) by TopHat. Calculation of gene expression values and differential expression analysis were performed using Cufflinks. Significance analysis of RNA-seq data was used to identify those genes significantly up- or down-regulated by treatments, with a FDR<0.05. Fold change was computed with average transcript levels compared to DMSO control values, which was in turn log$_2$-transformed and computed for Spearman correlation coefficients between samples. Gene ontology analysis was performed using the online "agriGO" tool (http://bioinfo.cau.edu.cn/agriGO/), and a heat map was created using the R "pheatmap" package.

**In vivo stability of compounds**. One-week-old Col-0 plants grown on half-strength MS agar media were transferred into half-strength MS liquid media and grown for another two days. Chemical stock solutions (10 mM in DMSO) were diluted into growth media to a final concentration of 10 μM and 3 h later, plants were washed three times using the same media. Then the plants were grown in one-quarter-strength MS liquid media and were sampled at 0, 12, 24, 36, 48, 72, and 96 h after the chemical treatment. The amount of chemicals was determined using UPLC-TripleTOF 5600+ at the Proteomics and Metabolomics Core Facility in the Shanghai Center for Plant Stress Biology (PSC), Chinese Academy of Science.

**Phenotypic assays**. A germination assay was carried out as follows. Seeds of Col-0 and the PYL triple mutant (*pyr1;pyl1;pyl4*) were stratified for 4 days before they were sown on half-strength MS solid medium containing 1% sucrose and 1 μM of the indicated compounds, with 15 seeds per line per 6 cm plate and four plates for each chemical. The plates were kept in a growth chamber at 22 °C under long-day conditions. Seeds were evaluated daily and were considered germinated when the green cotyledons appeared. A 0.05% solution of DMSO was used as the control for all chemicals.

Wild-type and p*RD29a-PYL2* transgenic *A. thaliana* and soybean plants were used in transpiration and drought resistance assays. Four-day-old *Arabidopsis* plants of identical size grown on half-strength MS solid medium were transferred to soil and grown under short-day conditions for another 10 days. Plants were then subjected to water withholding and were sprayed with chemical solutions, which contained 10 or 20 μM solutions of the indicated compounds and 0.05% Tween 20, once per week. For soybeans, plants at the triple trifoliate stage were subjected to water withholding and sprayed every 3 days with a 50 μM solution of the indicated compound and 0.1% Tween 20. A 0.05% solution of DMSO was used as the control for all chemicals. Each pot had the same amount of soil, and the position of pots and plates was changed every other day to minimize position effects. For leaf area measurements, 8-day-old soybean plants were subjected to water withholding and 3 and 8 days later were sprayed with DMSO or a 20 μM AMF4 solution. During the treatment, the plants were photographed by an image system every day, and the leaf area was calculated using OpenCV program. For the well-watered condition, the plants were irrigated every 3 days; for the drought stress condition, the plants were not irrigated until 22 days after water withdrawal. For *Arabidopsis*, one-month-old ABA-deficient mutant *aba2-1* was used in the transpiration assay to minimize the influence of endogenous ABA. Plants were sprayed with indicated compounds or (+)-ABA and photographed with an IR imager (FLIR) before and after chemical spray, and the transpiration rate was indicated by the leaf temperature.

**Data availability**. The authors declare that all the data supporting the findings of this study are available within the paper and its supplementary information files. The data sets generated or analyzed during the current study are available from the corresponding author on reasonable request. X-ray structure data are deposited in the RCSB Protein Data Bank (www.pdb.org). PBD codes are as follows PYL2–AMF1β–HAB1—5VRO, PYL2–AMF2α–HAB1—5VS5, PYL2–AMF1α–HAB1—5VR7, PYL2–AMF2β–HAB1—5VSQ, PYL2–AMF4–HAB1—5VSR, PYL2–AMC1β–HAB1—5VT7. RNA-seq data are deposited in NCBI's Gene Expression Omnibus (GEO) server and are accessible through the GEO Series accession number GSE101488.

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

## Acknowledgements

We thank Drs. Hua-Zhong Shi and Ashish Srivastava for critical reading of the manuscript. This work was supported by funding from the Chinese Academy of Sciences including the Science and Technology Service Network Initiative project KFJ-SW-STS-143.

## Author contributions

J.-K.Z. and H.E.X. conceived the research; M.-j.C. designed and performed most biochemical and physiological experiments; Y.-L.Z. designed and synthesized the chemicals; X.L. performed the AlphaScreen assay and protein crystallization; H.H. and W.-l.W. analyzed the RNA-seq data; X.E.Z. and J.D. analyzed the X-ray crystal structure data; W.-w.W. and A.Z. performed soybean drought stress assay; C.-z.Z. and T.S. performed soybean leaf area measurement; F.-x.W. provided technical assistance; M.-j.C. drafted, and J.-K.Z. revised and finalized the manuscript.

## Additional information

**Competing interests:** The authors declare no competing financial interests.

