## [Peer Review File · Nature Communications]

Reviewers' comments:

Reviewer #1 (Remarks to the Author):

The manuscript entitled "Combining chemical and genetic approaches to increase drought resistance in plants" reports on improved design of an existing ABA agonist, AM1, by the incorporation of halogens at aromatic ring positions. This additional charge increases the binding affinity and potency of the improved compounds known as AMFs. This is demonstrated *in silico* via docking studies and by functional assays of PYL2-HAB1 phosphatase binding and activity. Treatment of plants with AMFs also demonstrated a benefit in increased tolerance to drought with AMFs alone or in combination with PYL ABA receptor overexpression in transgenic plants.

The main conclusions of the manuscript are:

1. Based on the chemical structure of AM1 which lacks a carboxyl at its methylphenyl group compared to ABA, the authors added a series of fluorines to this ring to more closely mimic ABA charge distribution. The prediction was that this would increase binding affinity of AMFs for the PYL2 binding pocket via increase hydrogen binding.
2. PYL binding affinity was assessed by luminescence and HAB1 (PP2C phosphatase) by a phosphatase inhibition assay. Binding affinities for AM1s were greater than those for ABA or a chlorinated AM1 (AMC1 beta). The trend indicated that increased fluorination led to increased binding affinity. Two soybean PYLs were also examined for AMF binding and found to be of greater affinity compared to Arabidopsis PYLs. ABA bound to 11 tested PYLs and PYR1. AMFs showed strong inhibition of HAB1 activity for 6 of the 11 receptors and weaker HAB1 inhibition for PYLs 4, 6, 8, 9, and 10.
3. To examine the binding of AMFs in more detail, the crystal structure of PYL2-AMFs-HAB1 were generated and compared to those for complex with AM1. A combination of reduced hydrogen bond length differences and steric hindrance (among other changes documented in the manuscript) resulted in tighter binding to the hydrophobic pocket of PYL2. This was a significant effort.
4. RNASeq was used to examine the profiles of three week old plants treated with 10uM ABA and AMF1 beta and AMF4. ABA responsive genes showed a reduction after 24 hrs, whereas a significant number of such genes persisted at 72 hours post-treatment with AMF4, the most potent of the AMFs.
5. AMFs 2 alpha and AMF4 inhibited seed germination similar to ABA. AMF1 beta and AM1 inhibited to a lesser degree.
6. AMF treated Arabidopsis plants were examined in the *aba2-1* ABA deficient mutant by IR for leaf temperature differences indicating increased stomatal closure. Two days after treatment ABA and AMFs showed increased leaf temperature. However after four days the AMF2 alpha and AMF4 in particular persisted compared to ABA or AM1. Increased doses (5uM compared to 1uM) resulted in greater leaf temperature especially for AMF4. Similar experiments were done using soybean leaves although at 20uM AMF4.
7. Arabidopsis plants were foliar treated with AMFs and subjected to drought. AMFs were clearly more tolerant. Soybeans were also treated with 50uM AMF2 alpha and AFM4. Plants recovered to a greater extent following water withdrawal as assessed by total leaf area.

AMF4 had little or no effect on leaf area in the absence of drought.

8. Arabidopsis and soybean plants expressing PYL2 under an ABA and drought responsive promoter were then tested after AMF treatments. It appears that AtPYL2 expression in Arabidopsis was increased when treated with ABA and AMFs indicating that the promoter is responsive. Plants treated with AMFs showed gene expression patterns that correlated with ABA treatment but persisted longer over a three day period. Applications for AMF4 in particular to transgenic plants resulted in survival after even 28 days with higher soil moisture indicative of less transpiration. Similar dramatic results were obtained with transgenic soybean. IR measurements of soybean plants showed increased leaf temperature and higher soil moisture. The results point to an enhanced effect of AMFs in a PYL2 overexpression background.

Comments:

1. Suppl Fig 1. Why were these two soybean PYLs chosen? Are they more significant than others or were they just available for example? Also X axes should be labeled as concentration in micromoles comparable to figure 2.
2. Figure 2. It is very interesting that AMFs were less effective in inhibiting HAB1 activity for PYLs 4, 6, 8, 9, 10. The binding affinities for these PYLs are not shown in panels a – d. Do they have reduced binding affinity? What is the biological significance of the apparent selectivity of the AMFs for the different receptors?
3. Also in Suppl Fig 4, AMFs are less effective than ABA in inhibiting seed germination. The PYR1/PYL1/PYL4 triple mutant was interesting. The mutant is resistant to ABA which makes sense considering that it binds to PYR1 and PYL2 (PYL4 binding is not shown for ABA (Fig 2a-d)). ABA also affects the HAB1 activity of PYR1, PYL1 and PYL4. AMFs have little effect on PYL4 modulated HAB1 activity (Fig 2e). If my reasoning is correct the expectation is that the triple mutant would be LESS resistant to AMFs since they bind to only two of the three receptors knocked out in the triple mutant. In fact that mutant appears more resistant in particular to AMF AMF2 alpha and AMF4. How do the authors rationalize this biologically? For example does it suggest that PYR1 and PYL1 are more important receptors biologically for seed germination than PYL4? Even for a strictly biotechnology application receptor selectively seems very relevant to designing new and better ABA agonists. This seems important as a potentially biologically relevant component of the manuscript and should be discussed along with points in Comment 2.
4. Suppl Fig 6. In my version, the legend or text does not refer to the figure fully. The legend has no reference to panels c or d. ArRAB18 is not explained, so readers cannot interpret the figure. In Methods, qPCR primer sequences are not shown. Also no primer efficiencies, primer concentrations, thermocycler conditions or source of qPCR reagents are given. This cannot be replicated by others without complete information. This should be standard for reporting qPCR methods.
5. Figure 5. Can these IR leaf temperatures be quantified as average leaf temperature for example? It is obvious for some compounds that leaf temperatures persist; however, for AMF1 alpha at hour days for example is the leaf temperature the same as ABA?
6. Suppl Fig 5a. What is the unlabeled column of plants? Is this also ABA treatment?
7. I don't know what the editorial policy is in this case, but I would gently suggest that it should not be necessary to refer to figures in the Discussion as this should be a synthesis of the meaning of the Results overall.

8. Related to Comment 7. The Discussion needs significant improvement. It is mostly a restatement of the Results. For example lines 351 to 405 are dispensable for the most part due to redundancy with Results. I believe that the major points in the Discussion are synthetic PYL ligands that were rationally designed, the suggestion that AMFs are less susceptible to degradation (stated in Discussion) and discussion of the value of the overexpression strategy. While the impact of drought resistance is significant and obvious, what does the data say otherwise that is biologically relevant. See comments 2 and 3 for example. At a minimum the Discussion could be much shorter by removing redundancy. But I hope the authors will give thought to taking the opportunity to tell the readers what the data means and its significance.

8. Suppl Fig 7. In my view this figure which looks at the transcriptome profile of transgenic Arabidopsis treated with AMFs adds little to the manuscript which already has many figures and a huge amount of data.

Overall the manuscript is well written, logical and understandable. The data appears to be of high quality and repetitions are noted and stats are explained. The manuscript represents a huge amount of effort. The findings in terms of increase drought tolerance are significant for the readership of NC. However the revisions are too extensive for acceptance at this time so I would suggest rejection at this time. However, I would encourage the authors to address the comments and consider resubmission.

Reviewer #2 (Remarks to the Author):

In this manuscript the authors developed a series of ABA receptor agonists (AMFs) based on former reported ABA analog AM1/quinabactin. The analysis of crystals of PYL2-AMFs-HAB1 complex gave an insight into the structure-activity relationship of these analogs. That is, introduction of fluorine atoms into the 4-methylbenzyl group introduces more or stronger hydrogen bonds between AMFs and PYL2 pocket than the one between AM1 and PYL2 pocket, which supports the result that AMFs show higher affinity for PYL receptors than AM1. The authors indicated that AMFs induce longer lasting expression of genes that involved in abiotic-stress resistance than ABA in Arabidopsis. Consistent with that, some AMFs, such as AMF2a and AMF4, showed superior effects than ABA on protecting plants from drought stress in Arabidopsis and soybean. By combining the chemicals with stress-inducible overexpression of PYL2 (pRD29a:PYL2), drought resistance induced by AMFs was further improved. In summary, the authors developed new ABA analogs that are superior to ABA, and gave a new strategy for combating the drought stress.

The authors should give attention to the following points and suggestions.

1. The authors mentioned that "The EC50 values also indicated that the binding affinities to the four tested PYL receptors were lower for AMC1 β than for the AMFs" in Line 132-133 in manuscript. This description is not proper since AMC1 β shows lower EC50 (621.2 nM) than AMF1 α (783.7 nM), AMF1 β (2760 nM) and AMF2 β (640 nM), which indicates that AMC1 β has an equal or relative higher affinity to the PYL2 than these three AMFs.

2. The structure-activity relationship studies indicated that Asn173 is one of important residues for interaction of PYL2 with ABA and AMFs. It interesting to make a site mutation at Asn173 with Ala or other amino acids to see whether the replacement of Asn abolishes or increases the activities of ABA or AMFs on promoting the interaction between PYL2 and HAB1 or inhibiting the HAB1 activity.
3. The result of seed germination test in Supplemental Figure 4 is better to be displayed with chart. The conclusion about these results seems not consistent with the former report in Cao et al., Cell Research (2013) that seed germination inhibited by ABA (1 μ M) and AM1 (1 μ M) was completely recovered in *pyr1;pyl1;pyl4*.
4. Replacement of "Fig. 2a" in Line 386 with "Fig. 2c" or "Fig. 2" is better.
5. It is better to define the (A) and (B) that follow the amino acid residues in Figure 3 by adding (A)and (B) following the words "pocket" and "HAB1" respectively in the Line 635. Should "*pyr1;pyl1;pyl2*" in Line 750 be "*pyr1;pyl1;pyl4*"?
6. The description that "Col-0 wild-type plants are sprayed with..." in Line 644 should be "Col-0 wild-type plants are sprayed with..."
7. Stability of compounds often affects the activity of compounds. The authors should compare the stability of AM1 with that of other non-fluorinated AMFs.

Reviewer #3 (Remarks to the Author):

The manuscript entitled "Combining chemical and genetic approaches to increase drought resistance in plants" is a continuation of a chemical biology screen to identify new agonists of the ABA receptor. Previously, two groups had identified a compound AM1/quinbactin that acts as an ABA agonist for receptors that have functions in vegetative development. In particular, AM1 application improves drought tolerance in a number of plant species.

In this study the authors looked at the published ABA receptor structure for one of the receptors PYL2 and designed analogs of AM1 that may function better with respect to binding to the receptor. They are successful in the development of an AM1 analogs and further characterize their best one, AMF4. The work mostly revolves around showing AMF4 act much like ABA at the level of the receptor, ABA induced gene expression and most importantly improving drought tolerance. The authors do many requisite experiments to prove their point. None of these tests are novel and are the types of experiments one would expect. I would have liked to have seen an actual graph comparing ABA induced gene expression to AMF to get a real shape of the chemical inductions. The authors find an R value of ~ 0.75 , which is a bit low compared to what you might expect for a specific ABA agonist. This may reflect the ability of AMF to stick around longer than ABA, although the time frame for this experiment was 6 hrs.

The key test, for me in this manuscript was that premise the authors are trying to improve

AM1 beyond what is published so as to develop drought protectants for agriculture. This is a valid approach that is often used by Pharma to take a lead chemical and turn it into a drug. The authors do show AMF4 is improved over ABA with respect to improving drought tolerance by a number of tests. However, I found the improvements were not really that impressive. All drought experiments were done at concentrations of AMF4 that are similar to ABA (10-20 μ M). I was expecting the improvement to be 2-3 orders of magnitude. For this approach to have any success we will need compounds that act in the nanomolar or lower range and in this case the authors are well above that. Because the previous publications on AM1 I did not find the finding that exciting and it looks like improving these types of structures to a level where Ag-chem would be interested is still far away.

Reviewer #4 (Remarks to the Author):

Structural elucidation of abscisic acid (ABA) receptors has enabled us to develop agrochemicals that mimic ABA functions, especially enhancement of drought stress tolerance. Quinabactin, also referred to as AM1 in this article, is the most effective ABA agonist because it simultaneously activates five types of ABA receptors (PYR1, PYL1, PYL2, PYL3 and PYL5 in Arabidopsis). This article reports the fluorinated compounds of AM1, namely AMFs, which shows higher binding affinity to some ABA receptors than ABA and activates six types of ABA receptors including PYL7. As the most interesting feature, AMFs exert longer persistence of drought stress tolerance than ABA and AM1. This effect is also enhanced in the transgenic Arabidopsis and soybean plants with drought stress-inducible PYL2 overexpression. Although AMFs seem to be useful as agrochemicals that effectively induce drought stress responses, the plausible molecular mechanism is not explained by the provided data. The authors should revise the manuscript according to the following comments.

1. The EC₅₀ value of ABA toward PYL2-HAB1 interaction is much higher than the previously reported value (Okamoto et al. PNAS, 2013). In order to confirm the improved affinities of AMFs, the EC₅₀ value of AM1 should be measured using the same AlphaScreen assay and compared with the values of AMFs. The differences of binding modes between AM1 and AMFs should be also discussed on the basis of such a data set.

2. Provide the table summarizing the data collection and refinement statistics for X-ray crystallography. The information of preliminary validation report is not enough to judge the quality of crystal structures. In addition, the authors should present 3D structure images to overview the spatial differences between the binding modes of AM1 and AMFs. Ligplot is not useful to understand the reason why Arg83 forms interaction with the sulfonyl oxygen of AMFs instead of Lys64 in the binding of AM1.

3. Generally, fluorination is considered to increase the in vivo stability of compounds, which may be a major reason for the improved persistence of AMFs in drought stress tolerance rather than the increased binding affinity toward ABA receptors. The in vivo stability should be compared between AM1 and AMFs.

Point-by-point response to reviewer comments

Reviewers' comments:

Reviewer #1 (Remarks to the Author):

The manuscript entitled "Combining chemical and genetic approaches to increase drought resistance in plants" reports on improved design of an existing ABA agonist, AM1, by the incorporation of halogens at aromatic ring positions. This additional charge increases the binding affinity and potency of the improved compounds known as AMFs. This is demonstrated *in silico* via docking studies and by functional assays of PYL2-HAB1 phosphatase binding and activity. Treatment of plants with AMFs also demonstrated a benefit in increased tolerance to drought with AMFs alone or in combination with PYL ABA receptor overexpression in transgenic plants.

The main conclusions of the manuscript are:

1. Based on the chemical structure of AM1 which lacks a carboxyl at its methylphenyl group compared to ABA, the authors added a series of fluorines to this ring to more closely mimic ABA charge distribution. The prediction was that this would increase binding affinity of AMFs for the PYL2 binding pocket via increase hydrogen binding.
2. PYL binding affinity was assessed by luminescence and HAB1 (PP2C phosphatase) by a phosphatase inhibition assay. Binding affinities for AM1s were greater than those for ABA or a chlorinated AM1 (AMC1 beta). The trend indicated that increased fluorination led to increased binding affinity. Two soybean PYLs were also examined for AMF binding and found to be of greater affinity compared to Arabidopsis PYLs. ABA bound to 11 tested PYLs and PYR1. AMFs showed strong inhibition of HAB1 activity for 6 of the 11 receptors and weaker HAB1 inhibition for PYLs 4, 6, 8, 9, and 10.
3. To examine the binding of AMFs in more detail, the crystal structure of PYL2-AMFs-HAB1 were generated and compared to those for complex with AM1. A combination of reduced hydrogen bond length differences and steric hindrance (among other changes documented in the manuscript) resulted in tighter binding to the hydrophobic pocket of PYL2. This was a significant effort.
4. RNASeq was used to examine the profiles of three week old plants treated with 10uM ABA and AMF1 beta and AMF4. ABA responsive genes showed a reduction after 24 hrs, whereas a significant number of such genes persisted at 72 hours post-treatment with AMF4, the most potent of the AMFs.
5. AMFs 2 alpha and AMF4 inhibited seed germination similar to ABA. AMF1 beta and AM1 inhibited to a lesser degree.
6. AMF treated Arabidopsis plants were examined in the *aba2-1* ABA deficient mutant by IR for leaf temperature differences indicating increased stomatal closure. Two days after treatment ABA and AMFs showed increased leaf temperature. However after four days the AMF2 alpha and AMF4 in particular persisted compared to ABA or AM1. Increased doses (5uM compared to 1uM) resulted in greater leaf temperature especially for AMF4. Similar

experiments were done using soybean leaves although at 20uM AMF4.

7. Arabidopsis plants were foliar treated with AMFs and subjected to drought. AMFs were clearly more tolerant. Soybeans were also treated with 50uM AMF2 alpha and AFM4. Plants recovered to a greater extent following water withdrawal as assessed by total leaf area. AMF4 had little or no effect on leaf area in the absence of drought.

8. Arabidopsis and soybean plants expressing PYL2 under an ABA and drought responsive promoter were then tested after AMF treatments. It appears that AtPYL2 expression in Arabidopsis was increased when treated with ABA and AMFs indicating that the promoter is responsive. Plants treated with AMFs showed gene expression patterns that correlated with ABA treatment but persisted longer over a three day period. Applications for AMF4 in particular to transgenic plants resulted in survival after even 28 days with higher soil moisture indicative of less transpiration. Similar dramatic results were obtained with transgenic soybean. IR measurements of soybean plants showed increased leaf temperature and higher soil moisture. The results point to an enhanced effect of AMFs in a PYL2 overexpression background.

Comments:

1. Suppl Fig 1. Why were these two soybean PYLs chosen? Are they more significant than others or were they just available for example? Also X axes should be labeled as concentration in micromoles comparable to figure 2.

A1: Some PYLs can bind with PP2Cs in an ABA/ABA analog-independent manner, and thus are not suitable for the AlphaScreen assay because of high background. AtPYL1 and AtPYL2 can bind with HAB1 only in the presence of ABA. As indicated now in Lines 162-164, GmPYL3 and GmPYL6 are orthologs of AtPYL1 and AtPYL2, respectively (Bai et al, 2013, Line 164), which means that interactions of these two GmPYLs with HAB1 are also ABA/ABA analog-dependent and are suitable for the AlphaScreen assay. So we chose these two soybean PYLs to test if the synthetic ABA analogs may also work in soybean.

The X axes of Suppl Fig 1 are now labeled with "Conc (μ M)" as in Fig 2.

2. Figure 2. It is very interesting that AMFs were less effective in inhibiting HAB1 activity for PYLs 4, 6, 8, 9, 10. The binding affinities for these PYLs are not shown in panels a – d. Do they have reduced binding affinity? What is the biological significance of the apparent selectivity of the AMFs for the different receptors?

A2: As mentioned in Lines 75-79, PYLs 4, 6, 8, 9, 10 can bind with HAB1 in an ABA/ABA analog-independent manner and are thus not suitable for the AlphaScreen assay which we used to determine EC50, so that all the PYLs we chose in panel a-d show ABA/ABA analog-dependent binding with HAB1. For PYLs 4, 6, 8, 9, 10, the HAB1 activity assay in panel e shows that the AMF compounds are less effective than ABA at 1 μ M, which suggests reduced binding affinities of the AMFs to these PYLs. The apparent selectivity of AMFs for the PYLs implies that PYR1 and PYLs 1, 2, 3, 5, 7 share certain structural features that are different in PYLs 4, 6, 8, 9, 10. The PYL selectivity of AMFs may explain why the

AMFs were less effective than ABA in seed germination inhibition and why the *pyr1;pyl1;pyl4* triple mutant is more resistant to the AMFs than to ABA. Besides PYR1 and PYL1, some other PYLs that do not bind AMFs are also important for ABA-inhibition of seed germination. These points are now mentioned in the Discussion.

3. Also in Suppl Fig 4, AMFs are less effective than ABA in inhibiting seed germination. The PYR1/PYL1/PYL4 triple mutant was interesting. The mutant is resistant to ABA which makes sense considering that it binds to PYR1 and PYL2 (PYL4 binding is not shown for ABA (Fig 2a-d)). ABA also affects the HAB1 activity of PYR1, PYL1 and PYL4. AMFs have little effect on PYL4 modulated HAB1 activity (Fig 2e). If my reasoning is correct the expectation is that the triple mutant would be LESS resistant to AMFs since they bind to only two of the three receptors knocked out in the triple mutant. In fact that mutant appears more resistant in particular to AMF AMF2 alpha and AMF4. How do the authors rationalize this biologically? For example does it suggest that PYR1 and PYL1 are more important receptors biologically for seed germination than PYL4? Even for a strictly biotechnology application receptor selectivity seems very relevant to designing new and better ABA agonists. This seems important as a potentially biologically relevant component of the manuscript and should be discussed along with points in Comment 2.

A3: Thank you for raising this important point. The *pyr1;pyl1;pyl4* triple mutant seeds are more resistant to AMF2 alpha and AMF4 than to ABA. As the reviewer suggested, it is possible that PYL4 may not be as important as PYR1 and PYL1 in mediating inhibition of seed germination. Additionally, the greater resistance of the triple mutant seeds to AMF2 alpha and AMF4 may be because there are other PYLs that do not bind AMF2 alpha and AMF4 but are important for ABA-inhibition of seed germination. This is now discussed in the Discussion part.

4. Suppl Fig 6. In my version, the legend or text does not refer to the figure fully. The legend has no reference to panels c or d. ArRAB18 is not explained, so readers cannot interpret the figure. In Methods, qPCR primer sequences are not shown. Also no primer efficiencies, primer concentrations, thermocycler conditions or source of qPCR reagents are given. This cannot be replicated by others without complete information. This should be standard for reporting qPCR methods.

A4: The expression data on AtRAB18 and GmRAB18 contributes little to our conclusion and has been removed from this Figure. Details of qPCR are now included as suggested.

5. Figure 5. Can these IR leaf temperatures be quantified as average leaf temperature for example? It is obvious for some compounds that leaf temperatures persist; however, for AMF1 alpha at hour days for example is the leaf temperature the same as ABA?

A5: The average leaf temperatures have been quantified and added in Supplemental Fig 6. Four days post treatment, only plants treated with AMF2 beta or AMF4 still showed a

higher leaf temperature, and the leaf temperature of plants treated with other chemicals or ABA showed no difference with the DMSO control plants, as described in Lines 291-292.

6. Suppl Fig 5a. What is the unlabeled column of plants? Is this also ABA treatment?

A6: The unlabeled column of plants is now labeled "N/A" which indicates that they were treated with a chemical not described in this study.

7. I don't know what the editorial policy is in this case, but I would gently suggest that it should not be necessary to refer to figures in the Discussion as this should be a synthesis of the meaning of the Results overall.

A7: As suggested, we have removed reference to figures in the Discussion.

8. Related to Comment 7. The Discussion needs significant improvement. It is mostly a restatement of the Results. For example lines 351 to 405 are dispensable for the most part due to redundancy with Results. I believe that the major points in the Discussion are synthetic PYL ligands that were rationally designed, the suggestion that AMFs are less susceptible to degradation (stated in Discussion) and discussion of the value of the overexpression strategy. While the impact of drought resistance is significant and obvious, what does the data say otherwise that is biologically relevant. See comments 2 and 3 for example. At a minimum the Discussion could be much shorter by removing redundancy. But I hope the authors will give thought to taking the opportunity to tell the readers what the data means and its significance.

A8: We appreciate the comment, and have revised the Discussion as suggested.

9. Suppl Fig 7. In my view this figure which looks at the transcriptome profile of transgenic Arabidopsis treated with AMFs adds little to the manuscript which already has many figures and a huge amount of data.

A9: As the reviewer suggested, Suppl Fig 7 has been removed.

Overall the manuscript is well written, logical and understandable. The data appears to be of high quality and repetitions are noted and stats are explained. The manuscript represents a huge amount of effort. The findings in terms of increase drought tolerance are significant for the readership of NC. However the revisions are too extensive for acceptance at this time so I would suggest rejection at this time. However, I would encourage the authors to address the comments and consider resubmission.

Reviewer #2 (Remarks to the Author):

In this manuscript the authors developed a series of ABA receptor agonists (AMFs) based on former reported ABA analog AM1/quinabactin. The analysis of crystals of PYL2-AMFs-HAB1 complex gave an insight into the structure-activity relationship of these analogs. That is, introduction of fluorine atoms into the 4-methylbenzyl group introduces more or stronger hydrogen bonds between AMFs and PYL2 pocket than the one between AM1 and PYL2 pocket, which supports the result that AMFs show higher affinity for PYL receptors than AM1. The authors indicated that AMFs induce longer lasting expression of genes that involved in abiotic-stress resistance than ABA in Arabidopsis. Consistent with that, some AMFs, such as AMF2a and AMF4, showed superior effects than ABA on protecting plants from drought stress in Arabidopsis and soybean. By combining the chemicals with stress-inducible overexpression of PYL2 (pRD29a:PYL2), drought resistance induced by AMFs was further improved. In summary, the authors developed new ABA analogs that are superior to ABA, and gave a new strategy for combating the drought stress.

The authors should give attention to the following points and suggestions.

1. The authors mentioned that "The EC₅₀ values also indicated that the binding affinities to the four tested PYL receptors were lower for AMC1 β than for the AMFs" in Line 132-133 in manuscript. This description is not proper since AMC1 β shows lower EC₅₀ (621.2 nM) than AMF1 α (783.7 nM), AMF1 β (2760 nM) and AMF2 β (640 nM), which indicates that AMC1 β has an equal or relative higher affinity to the PYL2 than these three AMFs.

A1: Thank you for pointing out our mistake. We have re-done the assays for the AMF and AMC chemicals together with AM1, and revised the statement to more accurately describe the binding affinities of the tested PYLs for the chemicals.

2. The structure-activity relationship studies indicated that Asn173 is one of important residues for interaction of PYL2 with ABA and AMFs. It interesting to make a site mutation at Asn173 with Ala or other amino acids to see whether the replacement of Asn abolishes or increases the activities of ABA or AMFs on promoting the interaction between PYL2 and HAB1 or inhibiting the HAB1 activity.

A2: We thank the reviewer for the excellent suggestion. We have tested the binding affinities of AMF compounds to the mutated PYL2, and the result shows that the Asn173 to Ala mutation nearly abolished the activities of ABA or AMFs on promoting the interaction between PYL2 and HAB1 (Supplemental Figure 3).

3. The result of seed germination test in Supplemental Figure 4 is better to be displayed with chart. The conclusion about these results seems not consistent with the former report in Cao et al., Cell Research (2013) that seed germination inhibited by ABA (1 μ M) and AM1 (1 μ M) was completely recovered in pyr1;pyl1;pyl4.

A3: The seed germination result in Supplemental Figure 4 is actually quite consistent with

the former report in Cao et al., Cell Research (2013). Both results show that seed germination inhibited by ABA or AM1 was recovered in the pyr1;pyl1;pyl4 triple mutant. The difference is that the photograph in the Cao et al (2013) report was taken at a few days later compared to the one in our Supplemental Figure 4.

4. Replacement of "Fig. 2a" in Line 386 with "Fig. 2c" or "Fig. 2" is better.

A4: Per the suggestion of Reviewer 1, we have removed any reference to figures in the Discussion.

5. It is better to define the (A) and (B) that follow the amino acid residues in Figure 3 by adding (A) and (B) following the words "pocket" and "HAB1" respectively in the Line 635. Should "pyr1;pyl1;pyl2" in Line 750 be "pyr1;pyl1;pyl4"?

A5: We have added (A) and (B) following the words "pocket" and "HAB1" respectively in Line 635 (Line 651 in revised version). An overlay of 3-D schematics has also been added. The "pyr1;pyl1;pyl2" in Line 750 (Line 783 in revised version) should be "pyr1;pyl1;pyl4". We apologize for the typo.

6. The description that "Col-0 wild-type plants are sprayed with..." in Line 644 should be "Col-0 wild-type plants are sprayed with..."

A6: "Col-0 wild-type plants" in Line 644 (Line 665 in revised version) has been replaced with "Col-0 plants".

7. Stability of compounds often affects the activity of compounds. The authors should compare the stability of AM1 with that of other non-fluorinated AMFs.

A7: We appreciate the suggestion, and have compared the in-vivo stability of AMF4 with ABA and AM1. The result shows that AMF4 is more stable than AM1 and ABA (Lines 256-259 in revised version). Our results suggest that the physiological stability and high binding affinity both contribute to the longer lasting effect of AMF4.

Reviewer #3 (Remarks to the Author):

The manuscript entitled "Combining chemical and genetic approaches to increase drought resistance in plants" is a continuation of a chemical biology screen to identify new agonists of the ABA receptor. Previously, two groups had identified a compound AM1/quinabactin that acts as an ABA agonist for receptors that have functions in vegetative development. In particular, AM1 application improves drought tolerance in a number of plant species.

In this study the authors looked at the published ABA receptor structure for one of the receptors PYL2 and designed analogs of AM1 that may function better with respect to

binding to the receptor. They are successful in the development of an AM1 analogs and further characterize their best one, AMF4. The work mostly revolves around showing AMF4 act much like ABA at the level of the receptor, ABA induced gene expression and most importantly improving drought tolerance. The authors do many requisite experiments to prove their point. None of these tests are novel and are the types of experiments one would expect. I would have liked to have seen an actual graph comparing ABA induced gene expression to AMF to get a real shape of the chemical inductions. The authors find an R value of ~ 0.75 , which is a bit low compared to what you might expect for a specific ABA agonist. This may reflect the ability of AMF to stick around longer than ABA, although the time frame for this experiment was 6hrs.

The key test, for me in this manuscript was that premise the authors are trying to improve AM1 beyond what is published so as to develop drought protectants for agriculture. This is a valid approach that is often used by Pharma to take a lead chemical and turn it into a drug. The authors do show AMF4 is improved over ABA with respect to improving drought tolerance by a number of tests. However, I found the improvements were not really that impressive. All drought experiments were done at concentrations of AMF4 that are similar to ABA (10-20 μM). I was expecting the improvement to be 2-3 orders of magnitude. For this approach to have any success we will need compounds that act in the nanomolar or lower range and in this case the authors are well above that. Because the previous publications on AM1 I did not find the finding that exciting and it looks like improving these types of structures to a level where Ag-chem would be interested is still far away.

A: We respectfully disagree with the reviewer. The improvements achieved in this study is a result of improved binding affinities of the chemicals to PYLs, combined with increased physiological stability and increased sensitivity of the transgenic plants due to PYL overexpression. Although eventual large scale use by farmers may require further improvements, to our knowledge the drought resistance improvements achieved in this study represents the best reported thus far.

Reviewer #4 (Remarks to the Author):

Structural elucidation of abscisic acid (ABA) receptors has enabled us to develop agrochemicals that mimic ABA functions, especially enhancement of drought stress tolerance. Quinabactin, also referred to as AM1 in this article, is the most effective ABA agonist because it simultaneously activates five types of ABA receptors (PYR1, PYL1, PYL2, PYL3 and PYL5 in Arabidopsis). This article reports the fluorinated compounds of AM1, namely AMFs, which shows higher binding affinity to some ABA receptors than ABA and activates six types of ABA receptors including PYL7. As the most interesting feature, AMFs exert longer persistence of drought stress tolerance than ABA and AM1. This effect is also enhanced in the transgenic Arabidopsis and soybean plants with drought stress-inducible PYL2 overexpression. Although AMFs seem to be useful as agrochemicals that effectively induce drought stress responses, the plausible molecular mechanism is not explained by

the provided data. The authors should revise the manuscript according to the following comments.

1. The EC50 value of ABA toward PYL2-HAB1 interaction is much higher than the previously reported value (Okamoto et al. PNAS, 2013). In order to confirm the improved affinities of AMFs, the EC50 value of AM1 should be measured using the same AlphaScreen assay and compared with the values of AMFs. The differences of binding modes between AM1 and AMFs should be also discussed on the basis of such a data set.

A1: The Okamoto et al paper (PNAS) used PP2C inhibition assays to determine IC50, so the value cannot be directly compared with our EC50 data which was generated using the AlphaScreen assay, although the results from both assays showed the same trend when different chemicals or PYLs are compared. As an example of the two different assays giving very different values, in the Melcher et al paper (2009 Nature 462:602-608), the IC50 value of ABA towards PYL2-HAB1 interaction as determined by PP2C inhibition assay was 0.15 μ M, whereas the EC50 value of ABA towards PYL2-HAB1 interaction as determined by the AlphaScreen assay was about 5 μ M (Fig 1a and b in Melcher et al). As suggested by the reviewer, we have re-done the AlphaScreen assays to directly compare the binding affinities of AMFs and ABA with AM1. The new results as presented in revised Fig 2a-d show that all AMF compounds, but not AMC1 β , have higher binding affinities than ABA or AM1 to all four tested PYLs. The relative rank of affinities of the various AMFs and ABA in the new results is similar to that in the previous submission. However, EC50 values are very different between the new assay results and the previous submission. From our experience, the different batches of assay reagents (i.e. the beads for the fluorescence assay kit) can give very different EC50 values for the same chemicals and PYL-PP2C combinations, although the relative rank of binding affinities is not changed. Because the new assay results include AM1 and are thus more complete, and the greater EC50 values are more in line with those (e.g. low μ M range for ABA) in Cao et al (2013 Cell Res 23:1043-1054) and Melcher et al (2009 Nature 462:602-608) and the new results have been reproduced by multiple researchers, the new assay results are presented in the revised version. The GmPYL-AtHAB1 assays were also re-done with the new assay kit and the results (which show the same trend with data in the previous submission when the AMFs are compared with ABA) are included in revised Supplemental Fig.1.

2. Provide the table summarizing the data collection and refinement statistics for X-ray crystallography. The information of preliminary validation report is not enough to judge the quality of crystal structures. In addition, the authors should present 3D structure images to overview the spatial differences between the binding modes of AM1 and AMFs. Ligplot is not useful to understand the reason why Arg83 forms interaction with the sulfonyl oxygen of AMFs instead of Lys64 in the binding of AM1.

A2: We thank the reviewer for the helpful suggestions. The 3-D structure is now added in

Fig 3. Compared to AM1, the hydrogen bond between the fluorine atom of AMF4 and Asn-173 of PYL2 alters the relative position of AMF4 in the PYL2 pocket, which explains why the sulfonyl oxygen of AM1 and AMF4 forms hydrogen bond with different residues in PYL2.

3. Generally, fluorination is considered to increase the in vivo stability of compounds, which may be a major reason for the improved persistence of AMFs in drought stress tolerance rather than the increased binding affinity toward ABA receptors. The in vivo stability should be compared between AM1 and AMFs.

A3: We appreciate the suggestion, and have compared the in-vivo stability of AMF4 with ABA and AM1. The result shows that AMF4 is more stable than AM1 and ABA (Lines 256-259 in revised version). Our results suggest that the physiological stability and high binding affinity both contribute to the longer lasting effect of AMF4.

REVIEWERS' COMMENTS:

Reviewer #1 (Remarks to the Author):

This is a review of a resubmission of the manuscript titled "Combining chemical and genetic approaches to increase drought resistance in plants" that reports on improved design of an existing ABA agonist, AM1.

Point-by-point reviewer comments by the authors.

Thanks to the authors for addressing each of the comments and providing clear explanation. Overall the manuscript is much improved.

A few minor points:

1. Comment 1 Suppl Fig 1. Perhaps I missed it in the revised manuscript, but it seems worth noting in the manuscript that some PYLs can bind to PP2Cs in an ABA independent manner and not suitable for the Alpha screen and general readers who are not in the ABA field may find this useful.
2. Comment 4 qPCR. In Methods about line 573 Suppl Table 1 should be referenced where the primers are shown.
3. Comment 6. Thanks to the authors for clarifying. It is appreciated that the image was intact with an unrelated sets of plants. However, I would suggest that N/A be defined in the legend as plants from an unrelated experiment or the panel will still be confusing to readers. Alternatively those panels should be removed and a clear gap shown to indicate that the images were split. It could be noted in the legend that they were imaged at the same time. This comes down to editorial policy of Nat Comm.

Reviewer #2 (Remarks to the Author):

The manuscript is well corrected according to the comments except the response against the stability of AMF4. They demonstrated the stability of AMF4 in plant and likely think that the main reason of this stability of AMF4 may be due to the presence of fluorine atom. However they also mentioned in the text that the binding affinities of chemicals with PYL receptors were largely correlated with the number of fluorine atom. What is the main factor for the increase of the activity of fluorinated compounds, long lasting effect or increase of affinity to the receptor by introduction of fluorine atoms? I think they should perform more experiments to clarify this point or discuss more on this point.

Reviewer #4 (Remarks to the Author):

The authors have carefully revised the manuscript to solve my concerns. The additional data (Figs. 2a-d and Supplementary Figs. 1 and 5) show the improved EC50 values and in vivo stabilities of AMFs as compared with those of AM1. The 3-D structure image (Fig. 3) clearly presents the interaction between R83 and AMFs. However, the complete crystallographic

data should be provided to ensure the qualities of crystal structures as Supplementary Tables. The data should include the R_{pim} , $CC1/2$, $I/\sigma(I)$, completeness and redundancy in the overall resolution range and in the highest shell.

Point-by-point response to reviewer comments

Reviewer #1 (Remarks to the Author):

This is a review of a resubmission of the manuscript titled "Combining chemical and genetic approaches to increase drought resistance in plants" that reports on improved design of an existing ABA agonist, AM1.

Thanks to the authors for addressing each of the comments and providing clear explanation. Overall the manuscript is much improved.

A few minor points:

1. Comment 1 Suppl Fig 1. Perhaps I missed it in the revised manuscript, but it seems worth noting in the manuscript that some PYLs can bind to PP2Cs in an ABA independent manner and not suitable for the Alpha screen and general readers who are not in the ABA field may find this useful.

A1: We thank the reviewer for the kind reminder. In Lines 150-152, we did state that some PYLs can bind to PP2Cs in an ABA independent manner and are not suitable for the Alpha screen.

2. Comment 4 qPCR. In Methods about line 573 Suppl Table 1 should be referenced where the primers are shown.

A2: As suggested, we have now referenced Supplementary Table 1 in Lines 609-610 and 567-568 of the new version of our manuscript.

3. Comment 6. Thanks to the authors for clarifying. It is appreciated that the image was intact with an unrelated sets of plants. However, I would suggest that N/A be defined in the legend as plants from an unrelated experiment or the panel will still be confusing to readers. Alternatively those panels should be removed and a clear gap shown to indicate that the images were split. It could be noted in the legend that they were imaged at the same time. This comes down to editorial policy of Nat Comm.

A3: We thank the reviewer for the suggestions, and have now defined the "N/A" as "plants from an unrelated experiment" in Lines 974-975 in the legend for Supplementary Figure 8 (a).

Reviewer #2 (Remarks to the Author):

The manuscript is well corrected according to the comments except the response against the stability of AMF4. They demonstrated the stability of AMF4 in plant and likely think that the main reason of this stability of AMF4 may be due to the presence of fluorine atom. However they also mentioned in the text that the binding affinities of chemicals with PYL

receptors were largely correlated with the number of fluorine atom. What is the main factor for the increase of the activity of fluorinated compounds, long lasting effect or increase of affinity to the receptor by introduction of fluorine atoms? I think they should perform more experiments to clarify this point or discuss more on this point.

A: The reviewer asked a very good question. We do not know what is the main factor for the increase of activity of the fluorinated compounds. As mentioned in Lines 398-414, 428-430 and 437-440 in the Discussion, we believe that the increase of affinity to the receptors and long lasting effect together are responsible for the increase of the activity of the fluorinated compounds.

Reviewer #4 (Remarks to the Author):

The authors have carefully revised the manuscript to solve my concerns. The additional data (Figs. 2a-d and Supplementary Figs. 1 and 5) show the improved EC50 values and in vivo stabilities of AMFs as compared with those of AM1. The 3-D structure image (Fig. 3) clearly presents the interaction between R83 and AMFs. However, the complete crystallographic data should be provided to ensure the qualities of crystal structures as Supplementary Tables. The data should include the R_{pim}, CC1/2, I/σ(I), completeness and redundancy in the overall resolution range and in the highest shell.

A: As suggested, complete crystallographic data are now provided in Supplementary Table 2 and referenced in Lines 193-194 and 561-562. In addition, we have added an electron density map of PYL2-AMF4-HAB1 in Fig 3d, as required by the journal policy.